# A circuit-dependent ROS feedback loop mediates glutamate excitotoxicity to sculpt the *Drosophila* motor system

Jhan-Jie Peng[1,2†], Shih-Han Lin[1†], Yu-Tzu Liu[1], Hsin-Chieh Lin[1], Tsai-Ning Li[1], Chi-Kuang Yao[1,2]*

[1]Institute of Biological Chemistry, Academia Sinica, Taipei, Taiwan, Republic of China; [2]Institute of Biochemical Sciences, College of Life Science, National Taiwan University, Taipei, Taiwan, Republic of China

**Abstract** Overproduction of reactive oxygen species (ROS) is known to mediate glutamate excitotoxicity in neurological diseases. However, how ROS burdens can influence neural circuit integrity remains unclear. Here, we investigate the impact of excitotoxicity induced by depletion of *Drosophila* Eaat1, an astrocytic glutamate transporter, on locomotor central pattern generator (CPG) activity, neuromuscular junction architecture, and motor function. We show that glutamate excitotoxicity triggers a circuit-dependent ROS feedback loop to sculpt the motor system. Excitotoxicity initially elevates ROS, thereby inactivating cholinergic interneurons and consequently changing CPG output activity to overexcite motor neurons and muscles. Remarkably, tonic motor neuron stimulation boosts muscular ROS, gradually dampening muscle contractility to feedback-enhance ROS accumulation in the CPG circuit and subsequently exacerbate circuit dysfunction. Ultimately, excess premotor excitation of motor neurons promotes ROS-activated stress signaling that alters neuromuscular junction architecture. Collectively, our results reveal that excitotoxicity-induced ROS can perturb motor system integrity through a circuit-dependent mechanism.
DOI: https://doi.org/10.7554/eLife.47372.001

*For correspondence:
ckyao@gate.sinica.edu.tw

†These authors contributed equally to this work

Competing interests: The authors declare that no competing interests exist.

## Introduction

Reactive oxygen species (ROS) are generated as the by-product of mitochondrial oxidative phosphorylation (*Adam-Vizi, 2005*). In the central nervous system, under physiological conditions, high energy demand results in higher levels of ROS production relative to those in other body parts. In the past, endogenously generated ROS were recognized as signaling molecules that regulate a range of nervous system processes, including neuronal polarity, growth cone pathfinding, neuronal development, synaptic plasticity, and neural circuit tuning (*Li et al., 2016*; *Oswald et al., 2018a*; *Oswald et al., 2018b*). By contrast, ROS overproduction and/or overwhelming the antioxidant machinery can generate ROS burdens, termed oxidative stress, in aging and diverse pathological conditions (*Blesa et al., 2015*; *Bozzo et al., 2017*; *Liguori et al., 2018*; *Pollari et al., 2014*; *Wang et al., 2014*; *Zhao and Zhao, 2013*). In turn, excess ROS causes the malfunction and overactivation of ROS-regulated cell signaling pathways. Moreover, the highly oxidative properties of ROS are damaging to nucleotides, proteins, and lipids, eventually leading to neuronal dysfunction or demise. Hence, advancing our understanding of the mechanisms underlying ROS-induced neurotoxicity should aid the development of potent therapeutic treatments for neurological disorders.

Glutamate acts as the major excitatory neurotransmitter that regulates nearly all activities of the nervous system, with a tight balance between glutamate release and reuptake keeping the micromolar concentration of extracellular glutamate low (*Lewerenz and Maher, 2015*). In diseases, accumulation of extrasynaptic glutamate results in glutamate-mediated excitotoxicity to the nervous system

(*Dong et al., 2009*; *Mehta et al., 2013*; *Van Den Bosch et al., 2006*). Dysfunction of Na$^+$/K$^+$-dependent excitatory amino acid transporters (EAATs) is a key element of glutamate-mediated excitotoxicity (*Lewerenz and Maher, 2015*). In mammals, there are five EAAT subtypes, that is EAAT1 (GLAST), EAAT2 (GLT1), EAAT3 (EAAC1), EAAT4, and EAAT5 (*Vandenberg and Ryan, 2013*). EAAT3, EAAT4 and EAAT5 are expressed in neurons, whereas EAAT1 and EAAT2 are mainly present in astrocytes, where they are enriched in astrocyte terminal processes that form tripartite synapses with neurons and where they take up approximately 90% of released glutamate. Glutamate-mediated excitotoxicity can trigger bulk Ca$^{2+}$ influx into postsynaptic neurons via NMDA receptors, which causes mitochondrial Ca$^{2+}$ overload, along with other cellular responses, and which subsequently generates excess amounts of ROS (*Peng and Jou, 2010*; *Prentice et al., 2015*). Notably, it has emerged that dysregulation of neural circuit activity can initiate subsequent disruption of the integrity of other constituents in the same network, resulting in overall circuit dysfunction and even neurodegeneration (*Fornito et al., 2015*; *Hussain et al., 2018*; *Palop and Mucke, 2010*; *Shababi et al., 2014*). However, it still remains unclear whether and how excitotoxicity-induced ROS can influence the integrity of neural circuits.

Coordinated animal behaviors are linked to the activity of spinal cord central pattern generators (CPG), which are known to be specialized circuits that integrate inputs from the central brain and sensory neurons, and that subsequently generate rhythmic and patterned outputs to motor neurons (*Kiehn, 2016*). The *Drosophila* feed-forward locomotor circuit has served as an appropriate model for exploring the pathogenic network mechanisms that underlie neurodegenerative diseases (*Held et al., 2019*; *Imlach et al., 2012*; *Lotti et al., 2012*), because it has a relatively simple neural circuitry compared to mammals yet retains conserved functions (*Clark et al., 2018*; *Kohsaka et al., 2017*). In this study, we explored whether glutamate-mediated excitotoxicity impacts locomotor CPG activity, neuromuscular junction (NMJ) architecture, and motor function. Interestingly, we found that glutamate-mediated excitotoxicity due to depletion of *Drosophila* Eaat1, the sole *Drosophila* homolog of human EAAT2 (*Besson et al., 2000*), can induce a circuit-dependent ROS feedback loop that impairs the proper activities of the locomotor CPG circuit and muscles, ultimately leading to motor neuron overexcitation, abnormal NMJ growth and strength, and compromised movement. Together, our work reveals a circuit-dependent mechanism for increasing ROS, which mediates glutamate excitotoxicity to sculpt the *Drosophila* locomotion network.

## Results

### Loss of *Drosophila* Eaat1 causes motor-system deficits

Alterations in synaptic structure and function have been associated with a wide range of chronic neurodegenerative diseases (*Chand et al., 2018*; *Dachs et al., 2011*; *Fischer et al., 2004*; *Kariya et al., 2008*; *Lepeta et al., 2016*; *Rocha et al., 2013*; *Sharma et al., 2016*; *Wishart et al., 2006*). To explore the mechanisms underpinning these alterations, we collected the mutants that were previously characterized as having functional and/or developmental deficits in *Drosophila* photoreceptor cells (*Hiesinger et al., 2005*; *Jafar-Nejad et al., 2005*; *Ohyama et al., 2007*; *Verstreken et al., 2003*), and then conducted a secondary screen for mutations affecting synaptic bouton development of *Drosophila* third instar larval NMJs, with this latter system being broadly used to study the causes of neurodegenerative diseases (*Jaiswal et al., 2012*; *McGurk et al., 2015*). From this screen, we identified a hypomorphic mutation of *Drosophila* excitatory amino acid transporter 1 (*eaat1*). The mutant allele contained an insertion of a *roo* transposon in the last intron of the *eaat1* locus, which results in a severe reduction of Eaat1 protein levels (*Figure 1—figure supplement 1A–D*). Hereafter, we term this mutant *eaat1^hypo^*. Although *eaat1* null mutants (*eaat1^SM2/SM2^*) died before the first instar larval stage, as reported previously (*Stacey et al., 2010*), approximately 20% of *eaat1^hypo/hypo^* animals developed almost normally during the larval stages, but most of them died at early pupal stages (*Figure 1—figure supplement 1E*). We noted that a majority of *eaat1^hypo/SM2^* mutants died between the first and second instar larval stages, whereas ~1% of the mutants could further develop until the third instar stage, but they exhibited a significant ~10-day developmental delay (*Figure 1—figure supplement 1E*). The protein level of Eaat1 that remained in these survivors was further reduced relative to that detected in *eaat1^hypo/hypo^* mutants (*Figure 1—figure supplement 1C–D*).

These data indicate that the expression levels of Eaat1 are correlated with successful larval development.

To examine NMJ bouton architecture, we outlined the presynaptic and postsynaptic membranes of boutons by immunostaining with anti-horseradish peroxidase and anti-Disc large (Dlg) antibodies. As shown in *Figure 1A–D*, *eaat1^hypo* mutants displayed an ~50% increase in bouton number compared to wild-type controls, but their boutons were significantly smaller. Overall presynaptic area, active zone number per NMJ, and muscle surface area in controls and *eaat1^hypo* mutants were indistinguishable (*Figure 1—figure supplement 2A–C*). We then assessed whether synaptic transmission differs in response to morphological changes. Evoked excitatory junctional potential (EJP) was recorded from muscles under low-frequency (0.2 Hz) nerve stimulation. Compared to controls, *eaat1^hypo* mutants showed an increase in the EJP amplitude and quantal content (QC) (*Figure 1E–G*), whereas the amplitude and frequency of miniature EJPs were comparable between controls and *eaat1* mutants (*Figure 1H*; not shown for frequency), suggesting that loss of *eaat1* also abnormally increases neurotransmitter release.

*Drosophila* larval feed-forward locomotion is driven by the rhythmic activity of the locomotor CPG located in the ventral nerve cord (VNC) (*Cattaert and Birman, 2001*), reminiscent of human and mouse spinal cords. This premotor network integrates inputs from the central brain (*Cattaert and Birman, 2001*) and proprioceptive sensory neurons (*Hughes and Thomas, 2007*; *Song et al., 2007*), and it subsequently sends outputs to motor neurons (*Foran and Trotti, 2009*; *Shaw and Ince, 1997*). It has been shown that motor neurons that are derived from dissected first instar larvae of the *eaat1* null mutants receive sustained locomotor CPG output (*Stacey et al., 2010*). To corroborate this CPG phenotype, we generated third instar larval fillets of *eaat1^hypo* mutants, in which the brain and VNC remain intact, and performed intracellular muscle recordings to measure spontaneous locomotor CPG activity during fictive locomotion. Wild-type control motor neurons were normally able to receive rhythmic CPG output bursts that lasted a few seconds (*Figure 1I–J*), consistent with previous reports (*Cattaert and Birman, 2001*; *Imlach et al., 2012*). By contrast, *eaat1^hypo* motor neurons displayed a marked increase (~5 fold) in burst duration (*Figure 1I–J*). Moreover, burst frequency and intra-burst spike density were decreased in *eaat1^hypo* mutants relative to those in controls (*Figure 1—figure supplement 3A–B*). To represent the degree of sustained excitation on motor neurons quantitatively, we calculated the overall firing time of extremely long bursts (e.g. >15 s) for each recording minute and found that control and *eaat1^hypo* motor neurons were fired for 0.28 ± 0.28 s and 22 ± 5.38 s, respectively (*Figure 1K*). Consistent with these results, wild-type control larvae displayed coordinated peristalsis and locomotion activity, whereas *eaat1^hypo* mutants barely exhibited peristalsis and showed sluggish movement (*Figure 1L–M*). Therefore, these data suggest that Eaat1 is required to maintain proper patterning of the locomotor CPG output, and its loss can alter the CPG output pattern, thereby eliciting excess stimulation of motor neurons.

## Loss of *Drosophila* Eaat1 in astrocyte-like glia causes motor-system defects

*Drosophila* Eaat1 has been shown to be abundant in astrocyte-like glia of the central nervous system (*Besson et al., 2000*) (see also *Figure 2—figure supplement 1*). Although glutamate is the major neurotransmitter utilized by *Drosophila* motor neurons, Eaat1 is not present in the NMJ-associated glia throughout embryonic and larval development (*Rival et al., 2006*). To examine the physical association between the astrocyte-like glia and glutamatergic synapses in the larval VNC, we performed a GRASP (GFP reconstitution across synaptic partners) analysis (*Feinberg et al., 2008*), in which the transgenes of two split green fluorescence proteins (GFPs), *UAS-CD4::spGFP^1-10* and *lexAOP-CD4::spGFP^11*, were separately expressed in glutamatergic neurons and astrocyte-like glia using *OK371-GAL4* (also known as *vglut-GAL4*) and *repo-lexA*. Concurrently, the presynaptic compartments of glutamatergic interneurons were labeled by expressing the mCherry fusion transgene of an active zone scaffold protein Bruchpilot (Brp). We did not observe any GRASP signal in the VNCs derived from the larvae of either *OK371-GAL4/UAS-CD4::spGFP^1-10* or *repo-lexA/lexAOP-CD4::spGFP^11* control (*Figure 2—figure supplement 2*). By contrast, the larvae of *OK371-GAL4/UAS-CD4::spGFP^1-10/repo-LexA/lexAOP-CD4::spGFP^11* displayed high GRASP signals in neuropils of the dorsal part of the VNC (*Figure 2—figure supplement 2*; green in *Figure 2A*), which was tightly associated with Eaat1 expression (yellow in *Figure 2A*). However, we noted that the terminal

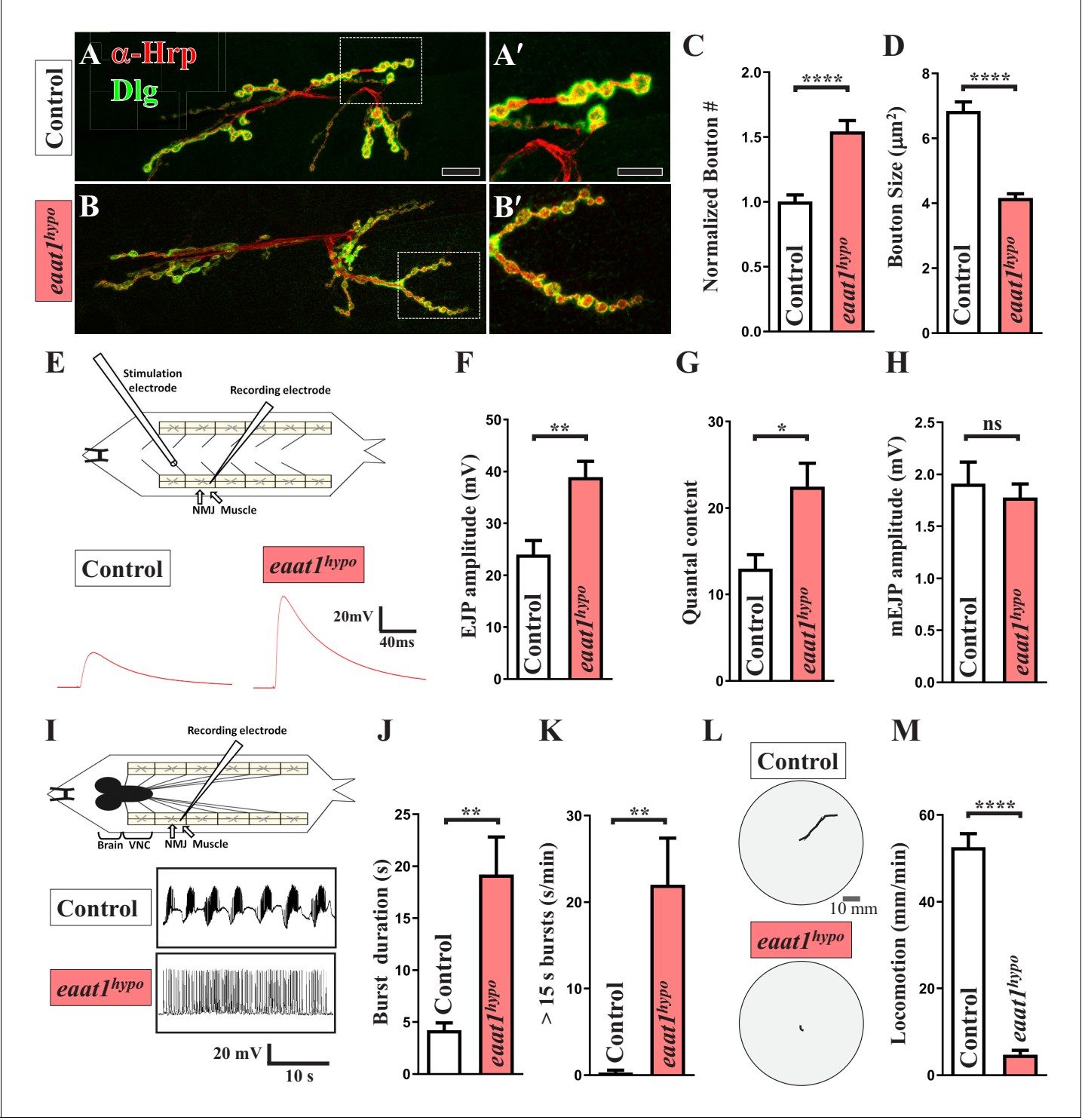

**Figure 1.** *eaat1* mutants exhibit NMJ bouton abnormalities, motor neuron overexcitation, and motor deficits. (**A–D**) Loss of *eaat1* increases NMJ bouton number and reduces bouton size. (**A–B**) Confocal images of NMJs co-stained with α-HRP (red) and α-Dlg (green) from controls (*w[1118]*) and *eaat1[hypo/hypo]* mutants. The NMJ boutons outlined in panels (**A,B**) are shown in panels (**A',B'**). Scale bars: 20 μm in (A,B), 10 μm in (A',B'). (**C**) The number of NMJ boutons per muscle area was counted and normalized to the value of controls (n ≥ 15 NMJs derived from A2 muscles 6 and 7 for each genotype). (**D**) The sizes of type Ib boutons were calculated on the basis of the immunostaining of Cysteine string protein, a synaptic vesicle-associated protein (n ≥ 433 type Ib boutons from NMJs (n ≥ 8) of A2 muscles 6 and 7 for each genotype). (**E–H**) Evoked presynaptic responses are increased upon loss of *eaat1*. (**E**) Top panel: schematic of the recording setting for larval fillets in which brain and ventral nerve cord (VNC) had been removed. Bottom panel: representative EJP traces evoked from A3 muscle 6 with 0.2 Hz electric stimulation in 0.5 mM Ca$^{2+}$-containing HL3 solution. (**F–H**) Quantification

*Figure 1 continued on next page*

*Figure 1 continued*

data for EJP amplitude, quantal content (QC), and miniature EJP amplitude (n ≥ 6 animals). Miniature EJPs were recorded in HL3 solution containing 0.5 mM $Ca^{2+}$ and 5 μM tetrodotoxin (TTX). (I–K) *eaat1^hypo* mutants receive excess premotor excitation. (I) Top panel: schematic of the recording setting for larval fillets in which the brain and VNC remain intact. Bottom panel: representative EJP traces evoked by spontaneous motor CPG activity during fictive locomotion. Recordings were obtained from A3 muscle 6 in HL3 solution containing 1 mM $Ca^{2+}$. (J–K) Quantification data for burst duration and overall firing time (from bursts of >15 s) per recording minute (n ≥ 6 animals). (L–M) *eaat1^hypo* mutants display compromised locomotion. (L) Representative locomotion tracks of third instar larvae. (M) Quantification data for larval locomotion (n ≥ 16 animals). P values: ns, no significance; *, p<0.05; **, p<0.01; ****, p<0.0001. n: replicate number. Error bars indicate the standard errors of the means (SEM). Statistics: Student's *t*-test.

DOI: https://doi.org/10.7554/eLife.47372.002

The following source data and figure supplements are available for figure 1:

**Source data 1.** Source data for *Figure 1*.
DOI: https://doi.org/10.7554/eLife.47372.009
**Figure supplement 1.** Characterization of the *eaat1^hypo* mutant allele.
DOI: https://doi.org/10.7554/eLife.47372.003
**Figure supplement 1—source data 1.** Source data for *Figure 1—figure supplement 1*.
DOI: https://doi.org/10.7554/eLife.47372.004
**Figure supplement 2.** Morphological analysis of *eaat1^hypo* mutant NMJ boutons and muscles.
DOI: https://doi.org/10.7554/eLife.47372.005
**Figure supplement 2—source data 1.** Source data for *Figure 1—figure supplement 2*.
DOI: https://doi.org/10.7554/eLife.47372.006
**Figure supplement 3.** Analysis of the properties of the locomotor CPG output.
DOI: https://doi.org/10.7554/eLife.47372.007
**Figure supplement 3—source data 1.** Source data for *Figure 1—figure supplement 3*.
DOI: https://doi.org/10.7554/eLife.47372.008

processes of astrocyte-like glia were close to rather than part of the glutamatergic tripartite synapses, which is consistent with a previous transmission electron microscopy study (*Stork et al., 2014*). To investigate the function of Eaat1, we expressed *UAS-iGluSnFR*, a membrane-bound extracellular glutamate sensor (*Chenji et al., 2016*; *Stork et al., 2014*), at glutamatergic synapses of the VNC using *OK371-GAL4* to measure the level of perisynaptic glutamate. As seen in *Figure 2B–C* (left panels), the control VNC exhibited low glutamate levels, whereas excess accumulation of glutamate was found upon Eaat1 depletion. Thus, Eaat1 maintains glutamate homeostasis efficiently through its close synaptic association in the *Drosophila* central nervous system.

To investigate whether the depletion of Eaat1 from astrocyte-like glia is causative of the observed motor system deficits, we first expressed the *UAS* transgene of the yellow fluorescent protein (Venus) fused to Eaat1 (*UAS-eaat1-venus*) in *eaat1^hypo* mutants using *repo-GAL4*, a pan-glia *GAL4* driver. This approach rescued the changes in locomotor CPG activity, NMJ boutons, and locomotion (*Figure 2D–I* and *Figure 1—figure supplement 3A–B*). In addition, a similar rescue effect was obtained with *alrm-GAL4*, an astrocyte-like glia-specific *GAL4* driver (*Doherty et al., 2009*) (*Figure 2D–I* and *Figure 1—figure supplement 3A–B*). Thus, loss of Eaat1 in astrocyte-like glia leads to the defective motor system phenotypes. Interestingly, when we also tested the effect of neuronal expression of *eaat1-venus*, altered locomotor CPG output activity and NMJ boutons, but not impaired locomotion, were rescued (*Figure 2D–I* and *Figure 1—figure supplement 3A–B*). This evidence suggests that the ectopic expression of Eaat1 in neurons can partially restore function, but appropriate expression of Eaat1 in astrocytes is required to coordinate CPG activity between and/or within individual segments. Next, we determined whether the functions of *Drosophila* Eaat1 are conserved in its human homolog. Expression of the *human EAAT2 (hEAAT2)-egfp* transgene using *repo-GAL4* rescued *eaat1^hypo* mutant phenotypes (*Figure 2D–I*), except for the reduced CPG burst frequency (*Figure 1—figure supplement 3A–B*). Under the above-described conditions, most rescued animals could eclose as adult flies (not shown). Therefore, the functions of EAAT are conserved from fly to human.

We further phenotypically characterized *eaat1^hypo/SM2* mutants that survived to the third instar stage. These larvae also exhibited alterations in locomotor CPG activity and locomotion (*Figure 2—figure supplement 3A–F*). In addition, they presented outgrown NMJ boutons (*Figure 2—figure supplement 3G–J*). However, *eaat1^hypo/SM2* mutant NMJs had numerous satellite boutons

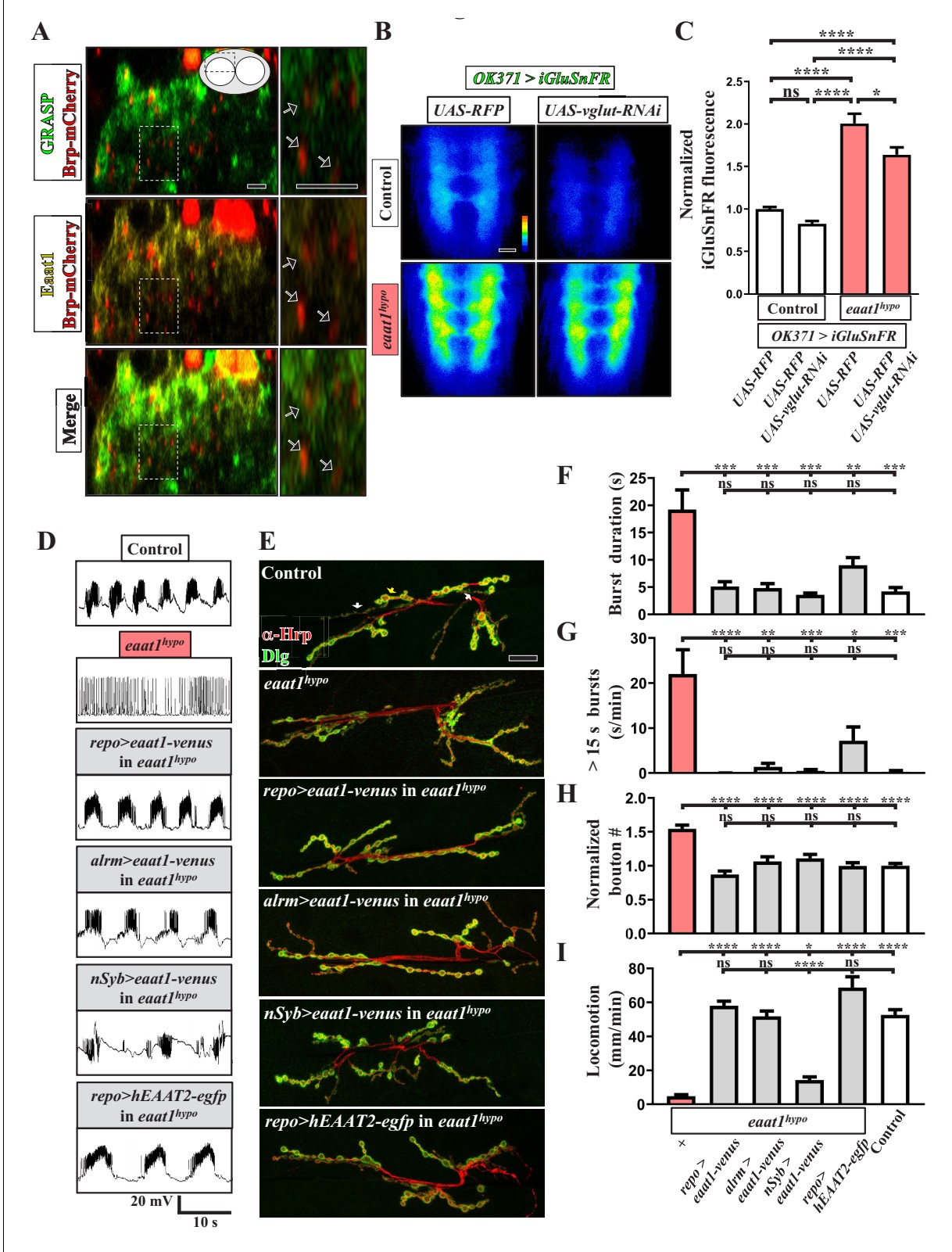

**Figure 2.** Eaat1 expressed in astrocyte-like glia plays a conserved role in maintaining motor system integrity. (**A**) GRASP assay showing a close association between astrocyte-like glia and motor neurons. Confocal cross-section images captured from the dorsal sector of the VNC of third instar larvae (*repo-LexA/LexAOP-CD4::spGFP[11]/OK371-GAL4/UAS-CD4::spGFP[1-10]/UAS-brp-mCherry*). Among the different types of glia, astrocyte-like glia predominantly extend their cellular processes into neuropils of the VNC. Arrows indicate clustered *Brp-mCherry*-labeled presynaptic compartments of

*Figure 2 continued on next page*

*Figure 2 continued*

glutamatergic interneurons, which are closely surrounded by GRASP signals (green) and Eaat1 proteins (yellow). Higher magnifications of the outlined regions are shown in the panels on the right. (B,C) Loss of *eaat1* elevates perisynaptic glutamate. (B) Pseudocolored images of the VNC-expressing iGluSnFR obtained from third instar larvae of the indicated genotypes. *UAS-RFP* was used as the control *UAS* transgene. Representative images were captured in zero calcium HL3 solution. (C) Quantification data for iGluSnFR signal intensity, normalized to the value of control (*OK371 >iGluSnFR +* RFP) (n ≥ 9 VNCs for each genotype). (D–I) Glial or astrocytic expression of the *eaat1-venus* transgene or of the *human EAAT2-egfp* transgene fully rescues the motor system defects in *eaat1^hypo^* mutants. Neuronal expression of the *eaat1-venus* transgene in *eaat1* mutants can restore premotor circuit activity and NMJ bouton growth but not locomotion. (D) Representative traces of EJPs evoked by spontaneous motor CPG activity during fictive locomotion obtained from third instar larvae of controls (*w^1118^*) and the indicated genotypes. Recordings were obtained from A3 muscle 6 in HL3 solution containing 1 mM $Ca^{2+}$. Quantification data for burst duration and overall firing time (for bursts of >15 s) for each recording minute are shown in panels (F,G) (n ≥ 6 animals for each genotype). (E) Confocal images of NMJs co-stained with α-HRP (red) and α-Dlg (green) obtained from third instar larvae of controls (*w^1118^*) and the indicated genotypes. Quantification data for NMJ bouton number for each muscle area, normalized to the value of controls, are shown in panel (H) (n ≥ 9 NMJs derived from A2 muscles 6 and 7 for each genotype). (I) Quantification data for the locomotion of third instar larvae of controls (*w^1118^*) and the indicated genotypes (n ≥ 10 animals for each genotype). P values: ns, no significance; *, $p < 0.05$; **, $p < 0.01$; ***, $p < 0.001$; ****, $p < 0.0001$. n: replicate number. Error bars indicate SEM. Statistics: one-way ANOVA with Tukey's post hoc test. Scale bars: 5 µm in (A), 20 µm in (B), 20 µm in (E).

DOI: https://doi.org/10.7554/eLife.47372.010

The following source data and figure supplements are available for figure 2:

**Source data 1.** Source data for *Figure 2*.
DOI: https://doi.org/10.7554/eLife.47372.015

**Figure supplement 1.** Expression pattern of Eaat1 in the larval VNC.
DOI: https://doi.org/10.7554/eLife.47372.011

**Figure supplement 2.** GRASP assay for the physical contact between astrocyte-like glia and glutamatergic neurons.
DOI: https://doi.org/10.7554/eLife.47372.012

**Figure supplement 3.** Phenotypic characterization of *eaat1^hypo/SM2^* mutants.
DOI: https://doi.org/10.7554/eLife.47372.013

**Figure supplement 3—source data 1.** Source data for *Figure 2—figure supplement 3*.
DOI: https://doi.org/10.7554/eLife.47372.014

(*Figure 2—figure supplement 3G–I*) and reduced muscle size (*Figure 2—figure supplement 3K*), outcomes that are distinct from those found in *eaat1^hypo/hypo^* mutants. We speculate that this morphological difference may be partly attributable to a significant developmental delay in NMJ bouton growth (*Figure 1—figure supplement 1E*) (*Sandoval et al., 2014*). Furthermore, glial expression of *eaat1-venus* using *repo-GAL4* robustly corrected these defects (*Figure 2—figure supplement 3F and I–K*). Hence, these results strengthen the causal role of the *eaat1* mutation in altering the integrity of the motor system.

## Glutamate-mediated excitotoxicity elicits premotor circuit dysfunction and motor neuron overexcitation upon loss of *eaat1*

In *Drosophila*, the locomotor CPG network activates motor neurons via cholinergic and GABAergic interneurons (Figure 4D, top panel) (*Rohrbough and Broadie, 2002*), and motor neurons do not receive direct excitatory glutamatergic inputs (*Kohsaka et al., 2017*). Moreover, *Drosophila* can utilize glutamate as an inhibitory neurotransmitter that activates glutamate-gated chloride channel α (GluClα) to elicit Cl⁻ influx (*Cully et al., 1996*). Glutamatergic period-positive median segmental interneurons (PMSIs) are responsible for relaying sensory inputs to motor neurons (*Clark et al., 2018*; *Kohsaka et al., 2014*). Nonetheless, removal of one copy of *gluClα* did not modify the motor-system defects associated with *eaat1^hypo^* mutants (*Figure 3—figure supplement 1*), raising the interesting possibility that excess synaptic glutamate caused by loss of *eaat1* may trigger excitotoxicity to alter locomotor CPG activity, which secondarily elicits excess stimulation of motor neurons. To test this possibility, we first neutralized the increased perisynaptic glutamate associated with *eaat1^hypo^* mutants by knocking down the *vesicular glutamate transporter* (*vglut*) in glutamatergic neurons, which is anticipated to reduce the glutamate loading of synaptic vesicles and hence glutamate release. Expression of *vglut-RNAi* using *OK371-GAL4* lowered excess perisynaptic glutamate in *eaat1^hypo^* mutants (*Figure 2B,C*). Furthermore, it largely rescued the altered locomotor CPG activity and impaired locomotion of *eaat1^hypo^* mutants (*Figure 3A–D*; *Figure 1—figure supplement 3A–B*). It has been shown previously that the locomotor CPG circuit is assembled within the VNC

(*Cattaert and Birman, 2001*). Consistently, when we expressed *UAS-eaat1-venus* solely in the VNC of *eaat1^hypo^* mutants using *teashirt(tsh)-GAL4*, it significantly reversed the prolonged CPG outputs (*Figure 3A–C*) and movement defects (*Figure 3D*). However, other CPG defects still remained (*Figure 1—figure supplement 3A–B*). Therefore, these results suggest that Eaat1 expressed in the VNC may modulate the locomotor CPG output pattern, especially burst duration, to protect motor neurons from excess premotor excitation. Eaat1 expressed in the central brain may be responsible for triggering the motor CPG and perhaps tuning intra-burst spike density. Therefore, the dual activities of Eaat1 control the speed and coordination of larval locomotion.

The AMPA-like glutamate receptor GluRIID is expressed in the central nervous system and is required for rhythmic locomotor CPG activity (*Featherstone, 2005*). Interestingly, like the expression of *eaat1-venus* driven by *tsh-GAL4*, removing one copy of *gluRIID* from *eaat1^hypo^* mutants reversed the prolonged CPG outputs (*Figure 3A–C*) but did not rescue the reduced burst frequency and intra-burst spike density (*Figure 1—figure supplement 3A–B*). However, *gluRIID* reduction only slightly benefited mutant locomotion (*Figure 3D*). Accordingly, it seems likely that excess glutamate could also affect coordination between segmental CPG networks by activating other types of glutamate-gated receptors, thereby leading to the defective locomotion. Together, these results suggest that loss of *eaat1* expression from the VNC leads to glutamate-mediated excitotoxicity, causing premotor circuit dysfunction and prolonged CPG output to motor neurons, and thereby gives rise to motor-system deficits.

## Oxidative stress elicits premotor circuit dysfunction upon loss of *eaat1*

It is known that glutamate excitotoxicity triggers bulk $Ca^{2+}$ influxes to promote mitochondrial $Ca^{2+}$ overload, overproducing ROS (*Zeeshan et al., 2016*). To investigate whether excess ROS production occurs within the locomotor CPG circuit upon glutamate excitotoxicity, we stained dissected VNCs with CM-H2DCFDA, a fluorescent ROS-sensing dye (*Nguyen et al., 2018*), to monitor cytosolic ROS levels. Compared to controls, increased ROS levels were evident in neuropils of *eaat1^hypo^* VNCs (*Figure 4A–B*). After abrogating excitotoxicity by reducing *gluRIID*, the increased ROS were significantly neutralized (*Figure 4A–B*), suggesting that glutamate-mediated excitotoxicity increases oxidative stress in the locomotor CPG circuit.

To investigate the effect of increased oxidative stress on locomotor CPG activity, we expressed the *UAS* transgene of human copper-zinc superoxide dismutase (hSOD1) in *eaat1^hypo^* mutants. This manipulation has been used previously to erase excess ROS and to rescue ROS-triggered neuronal defects in *Drosophila* (*Liu et al., 2015*; *Milton et al., 2011*). Pan-neuronal expression of *hSOD1* using *nSyb-GAL4* eliminated the increased ROS in *eaat1^hypo^* mutants (*Figure 4C*) and also rescued the defects in locomotor CPG activity and larval movement (*Figure 4D–G*). These results provide evidence that excess ROS in neurons contributes to premotor circuit dysfunction in *eaat1* mutants.

We then determined which population of neurons is potentiated and targeted by excess ROS. We first expressed *hSOD1* in glutamatergic neurons in *eaat1^hypo^* mutants using *OK371-GAL4*. However, neither the increased ROS within the VNC nor those within the motor-system phenotypes were reversed (*Figure 4C–G*). Furthermore, no rescue was observed when *hSOD1* was expressed in GABAergic neurons using *gad1-GAL4* (*Figure 4D–G*). Remarkably, expression of *hSOD1* in cholinergic neurons using *cha-GAL4* led to a significant restoration of the ROS level, locomotor CPG activity, and locomotion, comparable to the effect of pan-neuronal *hSOD1* expression (*Figure 4C–G*). Two subtypes of cholinergic sensory neurons, bipolar and class I, have been shown to deliver sensory feedback inputs to the locomotor CPG circuit during feed-forward locomotion (*Cheng et al., 2010*; *Hughes and Thomas, 2007*). However, targeted expression of *hSOD1* in both these neuron subtypes using *NP2225-GAL4* (*Imlach et al., 2012*) failed to affect the motor-system defects (*Figure 4D–G*). Hence, upon loss of *eaat1*, the excitotoxicity-induced ROS increase occurs in the cholinergic interneurons of the locomotor CPG network, dyregulating patterned locomotor CPG activity.

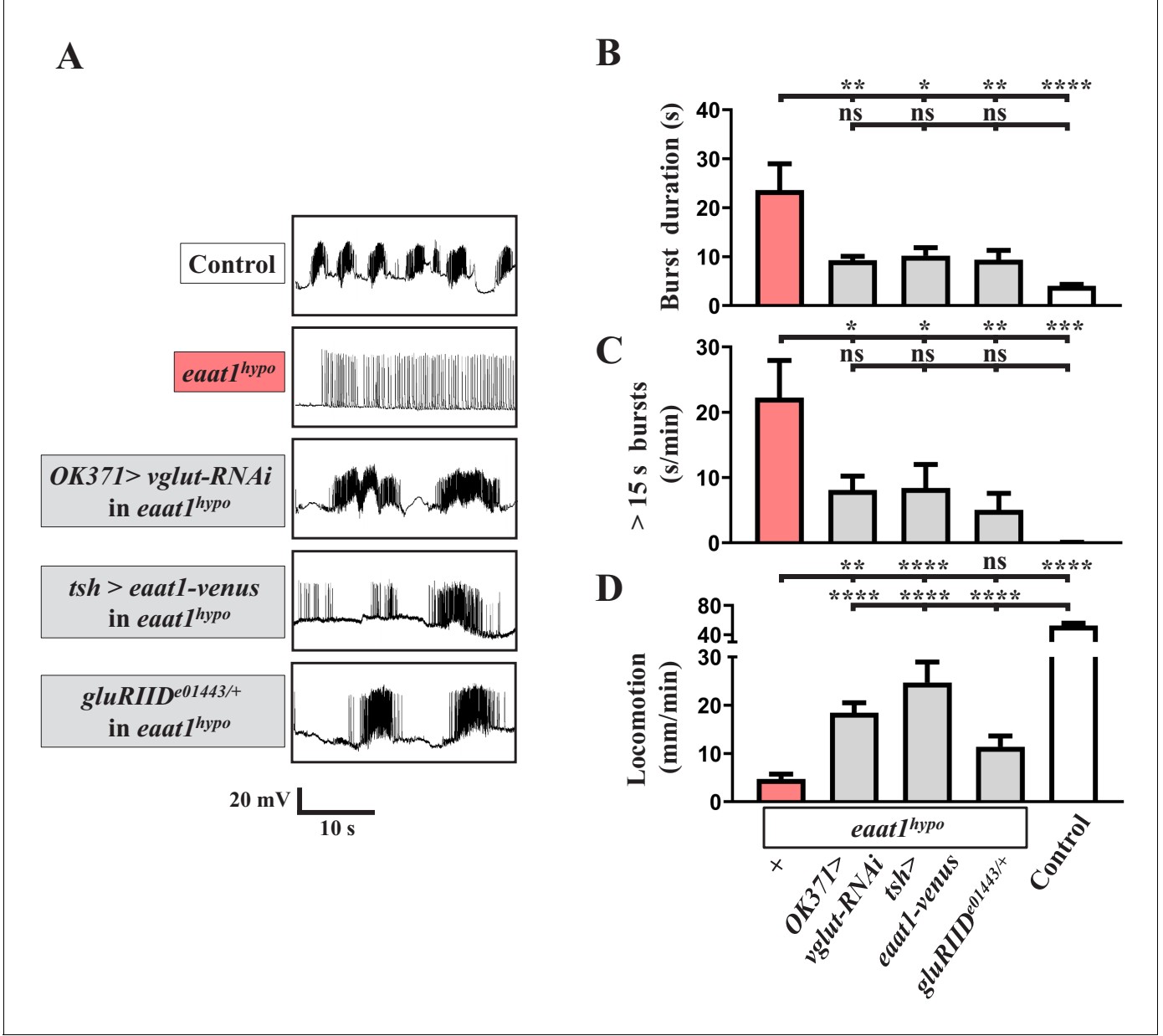

**Figure 3.** Glutamate-mediated excitotoxicity dysregulates premotor circuit activity in *eaat1* mutants. (A–D) Reducing glutamate release or inhibiting the postsynaptic glutamate receptor reverses prolonged burst duration in *eaat1^hypo* mutants. (A) Representative traces of EJPs evoked by spontaneous motor CPG activity during fictive locomotion obtained from third instar larvae of controls (*w^1118*) and the indicated genotypes. Quantification data for burst duration and overall firing time (for bursts of >15 s) per recording minute are shown in panels (B,C) (n ≥ 6 animals for each genotype). (D) Quantification data for locomotion of third instar larvae of controls (*w^1118*) and the indicated genotypes (n ≥ 11 animals for each genotype). P values: ns, no significance; *, p<0.05; **, p<0.01; ***, p<0.001; ****, p<0.0001. n: replicate number. Error bars indicate SEM. Statistics: one-way ANOVA with Tukey's post hoc test.

DOI: https://doi.org/10.7554/eLife.47372.016

The following source data and figure supplements are available for figure 3:

**Source data 1.** Source data for *Figure 3*.
DOI: https://doi.org/10.7554/eLife.47372.019
**Figure supplement 1.** Removal of one copy of *gluClα* does not affect the motor-system deficits caused by *eaat1* depletion.
DOI: https://doi.org/10.7554/eLife.47372.017
**Figure supplement 1—source data 1.** Source data for *Figure 3—figure supplement 1*.
DOI: https://doi.org/10.7554/eLife.47372.018

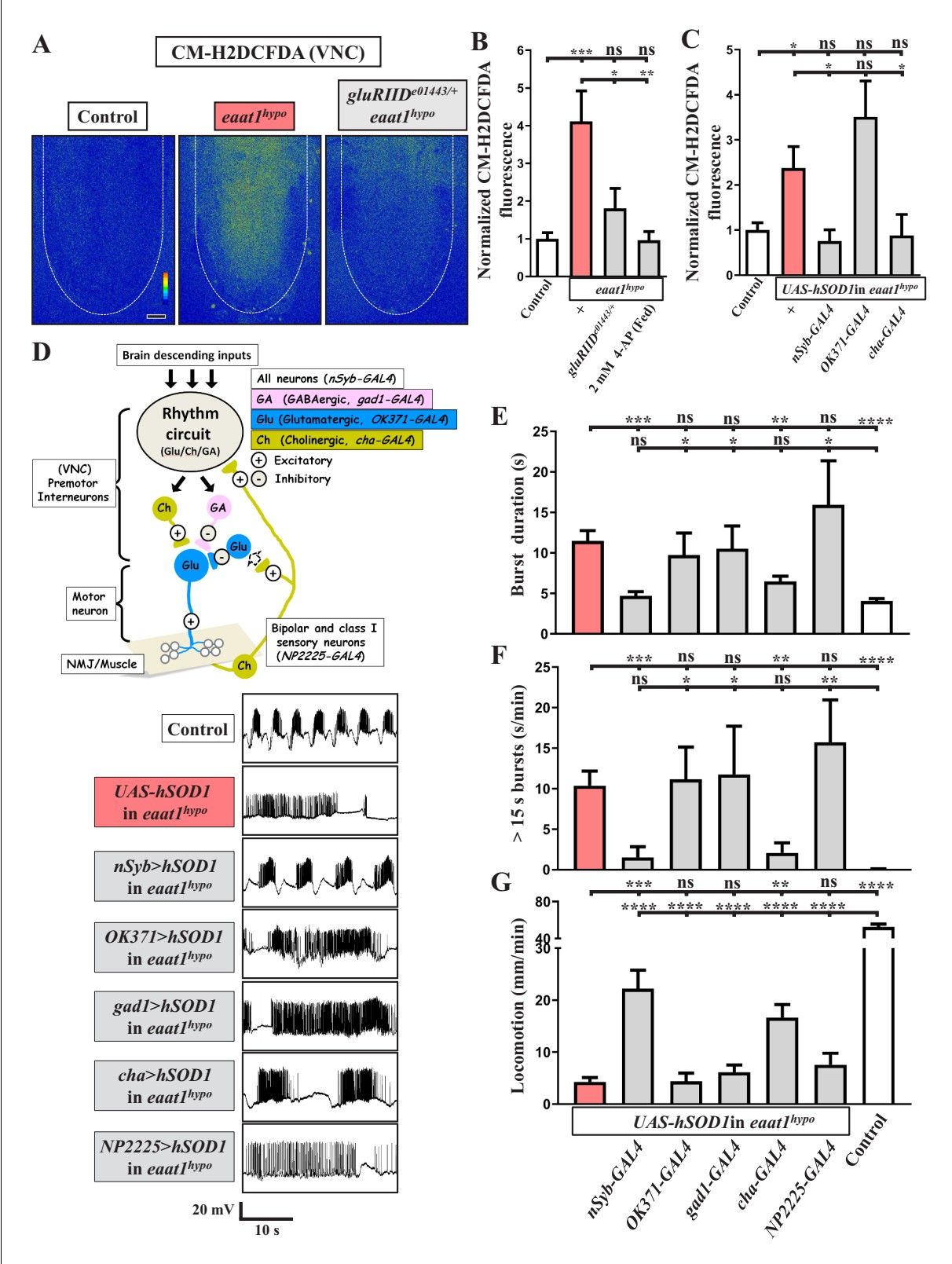

**Figure 4.** Loss of *eaat1* elevates oxidative stress in cholinergic interneurons, leading to premotor circuit dysfunction. (**A–C**) Loss of *eaat1* elevates ROS in the VNC through excess activation of glutamate receptors. (**A**) Pseudocolored confocal images of the VNCs (outlined with white dotted lines) stained with CM-H2DCFDA obtained from third instar larvae of controls (*w*[1118]) and the indicated genotypes. Scale bar: 20 µm. (**B**) Averaged CM-H2DCFDA fluorescence intensity in VNCs was normalized to the value of controls (n ≥ 6 VNCs for each genotype). (**C**) Averaged CM-H2DCFDA fluorescence

*Figure 4 continued on next page*

*Figure 4 continued*
intensity in VNCs of third instar larvae of controls (*w1118*) and the indicated genotypes, normalized to the value of controls (n ≥ 5 VNCs for each genotype). (D–G) Increased oxidative stress in cholinergic interneurons contributes to dysregulated premotor circuit activity. Schematic of neuronal type and connectivity in the *Drosophila* larval locomotor circuit (top panel). Corresponding neurons expressing specific *GAL4* drivers are indicated. Representative traces (bottom panel) of EJPs evoked by spontaneous motor CPG activity during fictive locomotion; these traces were obtained from third instar larvae of controls (*w1118*) and the indicated genotypes. (E–F) Quantification data for burst duration and overall firing time (for bursts > 15 s) per recording minute (n ≥ 6 animals for each genotype). (G) Quantification data for the locomotion of the third instar larvae of controls (*w1118*) and the indicated genotypes (n ≥ 10 animals for each genotype). P values: ns, no significance; *, p<0.05; **, p<0.01; ***, p<0.001; ****, p<0.0001. n: replicate number. Error bars indicate SEM. Statistics: one-way ANOVA with Tukey's post hoc test.
DOI: https://doi.org/10.7554/eLife.47372.020
The following source data is available for figure 4:

**Source data 1.** Source data for *Figure 4*.
DOI: https://doi.org/10.7554/eLife.47372.021

## Excitotoxicity-induced oxidative stress hampers the excitability of cholinergic interneurons upon loss of *eaat1*

Premotor circuit dysfunction due to Eaat1 depletion may be attributable to either abnormal network assembly or dysregulated network activity. The *Drosophila* larval locomotor CPG circuit is assembled during the late embryonic stage (*Kohsaka et al., 2012*). Accordingly, we utilized the *GAL4/GAL80^ts^* system to control the expression of *UAS-eaat1-venus* spatiotemporally in the *eaat1* mutant background. As shown in *Figure 5A,B*, the embryos of the control group (*repo-GAL4/UAS-eaat1-venus/ tub-GAL80^ts^* in *eaat1^hypo^* mutants) were reared at 18°C to restrict the expression of *eaat1-venus* from 2 hr after egg laying to the late larval stage. As expected, the developed third instar (L3) larvae behaved like *eaat1^hypo^* mutants. When these embryos were reared at 29°C to allow expression of *eaat1-venus* throughout embryonic and larval development, the L3 larvae exhibited full restoration of movement (termed 'full rescue'). However, restricted expression of *eaat1-venus* within the window of embryonic development (termed 'embryonic rescue') still resulted in defective locomotion. By contrast, the locomotion defect could be robustly normalized when the hatched larvae were reared at 29°C only during the larval stage (termed 'L1–L3 rescue'). Thus, *eaat1* is functionally required during the larval stage but not during embryonic development when the CPG circuit is assembled. Consistent with these results, we did not observe overt changes in VNC architecture in *eaat1^hypo^* mutants with regard to the number of neurons and astrocytes, the processes of astrocyte-like glia, and the dendritic field of motor neurons (*Figure 5—figure supplement 1*).

Intriguingly, when expression of *eaat1-venus* was switched on in *eaat1^hypo^* mutants progressively from L2, early L3, or even the middle L3 stage, all of these expression conditions provided robust rescue of locomotion (termed 'L2-L3', 'L3' and 'mL3' rescues, respectively, *Figure 5A–B*), suggesting that excitotoxicity-induced oxidative stress may result in temporal and reversible dysregulation of locomotor CPG activity, which can be normalized whenever the Eaat1 function is recovered. Accordingly, it is possible that acutely increasing oxidative stress in the locomotor CPG circuit would mimic the effect of the *eaat1* mutation. To test this possibility, we exposed dissected wild-type larval fillets to low concentrations of $H_2O_2$ (0.003% and 0.006%) for ~5 min and measured locomotor CPG activity. Remarkably, $H_2O_2$ exposure could prolong CPG output duration in a dose-dependent manner (*Figure 5C and F–G*). By contrast, when the anti-oxidative ability of cholinergic neurons was enhanced by expressing *hSOD1*, the circuit effect of $H_2O_2$ was greatly diminished (*Figure 5C and F–G*). Together, these data suggest that, upon loss of *eaat1*, a mild level of ROS does not perturb the synaptic connectivity of the locomotor circuit but instead alters its activity by influencing cholinergic transmission.

Oxidization of voltage-gated potassium channels caused by excess ROS can attenuate their inactivation, resulting in neuronal silencing under disease conditions (*Sahoo et al., 2014*; *Sesti et al., 2010*). Therefore, we hypothesized that premotor circuit dysfunction upon loss of *eaat1* may arise from ROS-mediated inactivation of cholinergic interneurons. The $K^+$ channel blocker 4-aminopyridine (4AP) is a clinical treatment for multiple sclerosis (*Chwieduk and Keating, 2010*). We incubated dissected wild-type larval fillets with 0.2 mM 4-AP-containing media for 5 min and then recorded CPG activity in the setting of 0% or 0.006% $H_2O_2$ treatment. Administration of 4-AP slightly increased the

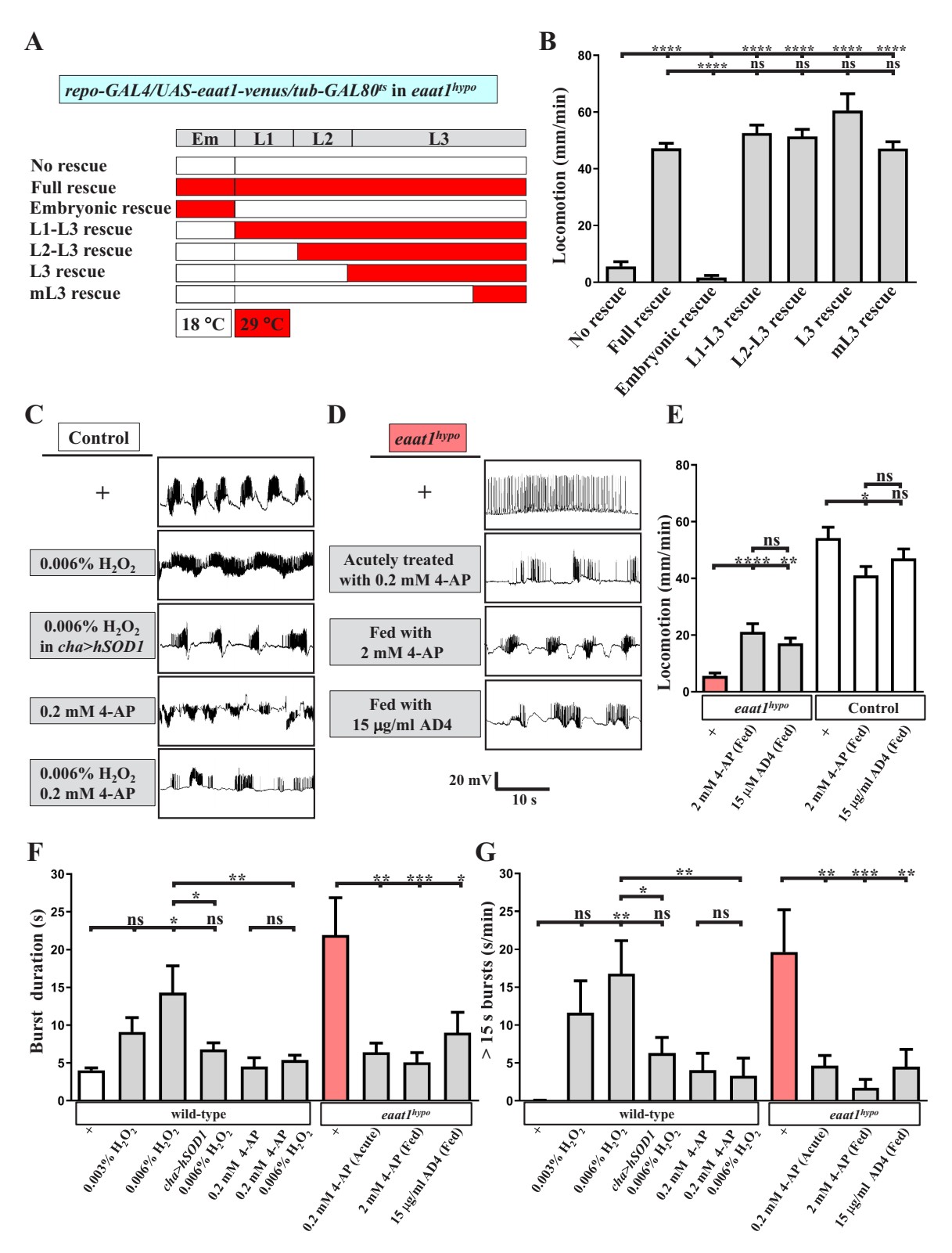

**Figure 5.** Increased excitotoxicity-induced ROS hampers the excitability of cholinergic interneurons. (A–B) The locomotion defect caused by loss of *eaat1* is rescued by expressing *eaat1-venus* at the late larval stage. (A) Schematic of the temporal expression of *UAS-eaat1-venus* controlled by the *GAL80^{ts}/GAL4* system during embryonic and larval stages. Animal genotypes are indicated. Developmental timescale of larvae at 18°C and 29°C: Em (embryo stage, 0–48 hr after egg laying (18°C) and 0–12 hr after egg laying (29°C)); L1 (first instar larvae, 0–48 hr after embryo hatching (18°C) and 0–12
*Figure 5 continued on next page*

Figure 5 continued

hr after embryo hatching (29°C)); L2 (second instar larvae, 48–96 hr after embryo hatching (18°C) and 12–24 hr after embryo hatching (29°C)); and L3 (third instar larvae, 96–240 hr after embryo hatching (18°C) and 24–60 hr after embryo hatching (29°C)). Expression of *eaat1-venus* was restricted at 18°C (white boxes), but was switched on at 29°C (red boxes). (B) Quantification data for locomotion of third instar larvae (n ≥ 8 third instar larvae for each genotype). (C) Acute exposure of $H_2O_2$ phenocopies the prolonged burst duration of the motor CPG, which is reversed by expression of *hSOD1* in cholinergic interneurons or by 0.2 mM 4-AP treatment. The representative traces show EJPs evoked by spontaneous motor CPG activity during fictive locomotion obtained from third instar wild-type controls (*w[1118]*) and third instar wild-type controls expressing *hSOD1* with *cha-GAL4*. Larval fillets were acutely exposed to 0.006% $H_2O_2$-containing HL3 solution for 3 min, followed by 10 min recordings in the same solution. For 4-AP treatment, larval fillets were bathed in 0.2 mM 4-AP-containing HL3 solution for 5 min and then bathed in 0.2 mM 4-AP/0.006% $H_2O_2$-containing HL3 solution for 3 min, followed by 10 min recordings in the same solution. Quantification data for burst duration and overall firing time (for bursts of >15 s) per recording minute are shown in panels (F,G) (n ≥ 8 animals for each genotype). (D) Prolonged burst duration caused by loss of *eaat1* can be rescued by acute treatment of 0.2 mM 4-AP or by long-term feeding of 2 mM 4-AP or 15 µg/ml AD4. Representative EJP traces evoked by spontaneous motor CPG activity during fictive locomotion obtained from third instar *eaat1[hypo/hypo]* mutants. Larval fillets were bathed in 0.2 mM 4-AP-containing HL3 solution for 5 min, followed by 10 min recordings in the same solution. For long-term drug treatment, *eaat1[hypo]* mutants were fed with 2 mM 4-AP or 15 µg/ml AD4 throughout the larval stage. Larval fillets were subjected to recordings in HL3 solution. Quantification data for burst duration and overall firing time (for bursts of >15 s) per recording minute are shown in panels (F,G) (n ≥ 7 animals for each genotype). (E) Long-term feeding of 2 mM 4-AP or 15 µg/ml AD4 improves locomotion of *eaat1[hypo]* mutants. 2 mM 4-AP but not 15 µg/ml AD4 slightly reduced locomotion of control (*[w1118]*) larvae. Locomotion of treated larvae was measured and quantified (n ≥ 17 animals for each genotype). P values: ns, no significance; *, p<0.05; **, p<0.01; ***, p<0.001; ****, p<0.0001. n: replicate number. Error bars indicate SEM. Statistics: one-way ANOVA with Tukey's post hoc test.

DOI: https://doi.org/10.7554/eLife.47372.022

The following source data and figure supplements are available for figure 5:

**Source data 1.** Source data for *Figure 5*.
DOI: https://doi.org/10.7554/eLife.47372.025
**Figure supplement 1.** No overt change in the structural integrity of the larval VNC upon loss of *eaat1*.
DOI: https://doi.org/10.7554/eLife.47372.023
**Figure supplement 1—source data 1.** Source data for *Figure 5—figure supplement 1*.
DOI: https://doi.org/10.7554/eLife.47372.024

CPG output duration in non-$H_2O_2$-treated animals (*Figure 5C and F–G*), presumably due to enhanced excitability of the locomotor circuit. Nonetheless, 4-AP administration reversed prolonged CPG outputs induced by 0.006% $H_2O_2$ treatment (*Figure 5C and F–G*). In addition, acute 4-AP treatment alleviated the CPG output defect associated with the *eaat1* mutation (*Figure 5D and F–G*). Next, we fed *eaat1[hypo]* mutants with food containing 2 mM 4-AP throughout the larval stages. This manipulation also abrogated the locomotion defect (*Figure 5E*). Our CPG recordings revealed that ~70% (12/17) of 4-AP-treated mutants displayed normalized CPG output patterns (*Figure 5D and F–G*), whereas the remaining animals (5/17) still showed mutant-like alterations (not shown). This incomplete rescue could be due to insufficient drug penetration across the blood–brain barrier surrounding the VNC. Last, treatment of *eaat1[hypo]* mutants with 15 µg/ml N-acetylcysteine amide (AD4), a potent membrane penetrating antioxidant (*Liu et al., 2015*), also significantly corrected both premotor circuit dysfunction and compromised locomotion (*Figure 5D–G*). Notably, when wild-type controls were fed with the food containing either 2 mM 4-AP or 15 µg/ml AD4, only a slight reduction in locomotion activity was observed (*Figure 5E*), indicating that the above-described rescue effects do not result from an overall increase in locomotor CPG activity. Taken together, these findings strongly suggest that the excitotoxicity-induced increase in ROS upon loss of *eaat1* hampers the excitability of cholinergic interneurons, leading to premotor circuit dysfunction.

## ROS-induced muscle weakness feedback exacerbates premotor circuit dysfunction upon loss of *eaat1*

Given the impaired locomotion phenotype, we then investigated whether excess excitation of motor neurons upon Eaat1 depletion could boost mitochondrial ROS production in larval muscles and influence muscular physiology. Mitochondrial ROS in muscles was probed by expressing mitotimer using *C57-GAL4*, a muscle-specific *GAL4* driver. Mitotimer is a mitochondria-targeted timer that switches its fluorescence from green fluorescent protein (GFP) to red fluorescent protein (RFP) upon oxidation (*Laker et al., 2014*). Compared to control muscles, the RFP/GFP ratio of mitotimer was significantly higher in *eaat1[hypo]* mutant muscles (*Figure 6A,B*), indicating an increased level of mitochondrial ROS. Note that the distribution of mitochondria in muscles is also slightly altered in

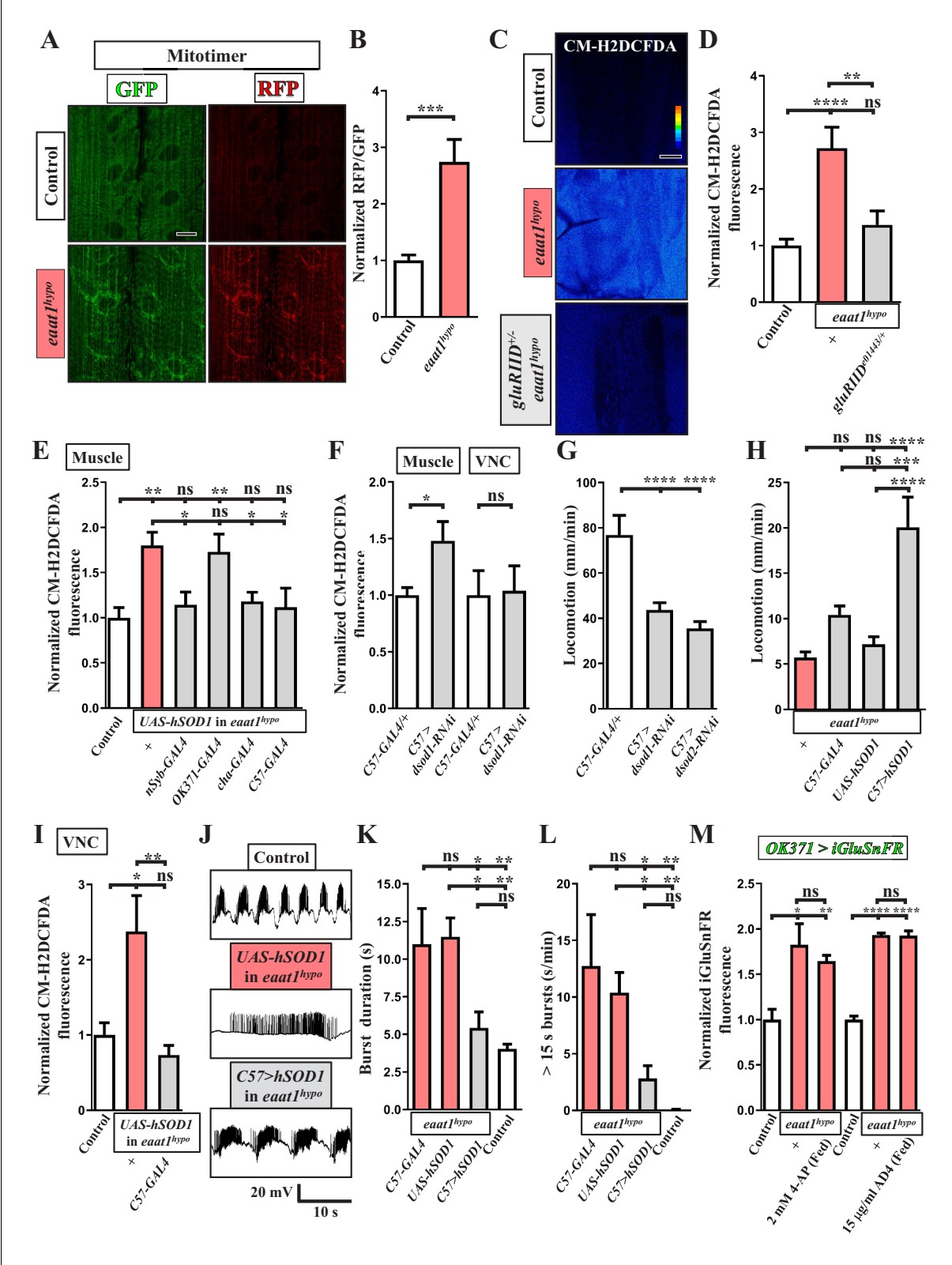

**Figure 6.** ROS-induced muscle weakness feedback maintains the increased premotor oxidative stress to exacerbate circuit dysfunction. (A,B) Loss of *eaat1* induces mitochondrial oxidative stress in muscles. (A) Confocal images of muscles 6 and 7 of third instar larvae expressing mitotimer using *C57-GAL4* obtained from wild-type controls (*w[1118]*) and *eaat1[hypo/hypo]* mutants. Scale bar: 20 μm. (B) The RFP/GFP ratio of mitotimer is assessed and normalized to the value of controls (n ≥ 6 animals for each genotype). (C,D) Reducing *gluRIID* can normalize the increased oxidative stress
*Figure 6 continued on next page*

Figure 6 continued

resulting from loss of *eaat1*. (C) Pseudocolored confocal images of muscles 6 and 7 of the third instar larvae of controls (*w1118*) and the indicated genotypes stained with CM-H2DCFDA. Scale bar: 20 μm. (D) Averaged CM-H2DCFDA fluorescence intensity was quantified and normalized to the value of controls (n ≥ 13 A3 muscles 6 and 7 from (n ≥ 7) animals for each genotype). (E) Expression of *hSOD* using *nSyb-GAL4*, *cha-GAL4* or *C57-GAL4* but not *OK371-GAL4* relieves the muscular oxidative stress of *eaat1hypo* mutants. CM-H2DCFDA fluorescence intensity of muscles 6 and 7 from the third instar larvae of controls (*w1118*) and the indicated genotypes is quantified and normalized to the value of controls (n ≥ 12 A3 muscles 6 and 7 from (n ≥ 7) animals for each genotype). (F) Muscular expression of *dsod1-RNAi* using *C57-GAL4* promotes ROS production in muscles but not in VNCs. Quantification data for CM-H2DCFDA fluorescence are shown. (n ≥ 7 animals for each genotype.) (G) Muscular knockdown of *dsod1* or *dsod2* impairs larval locomotion. *UAS-dsod1-RNAi* or *UAS-dsod2-RNAi* is expressed by using *C57-GAL4*, and larval locomotion was quantified (n ≥ 21 animals for each genotype). (H) Expression of *hSOD1* using *C57-GAL4* improves *eaat1hypo* mutant locomotion. Locomotion of third instar larvae of the indicated genotypes is quantified (n ≥ 23 animals for each genotype). (I–L) Expression of *hSOD1* using *C57-GAL4* relieves premotor oxidative stress of *eaat1hypo* mutants and reverses premotor circuit dysregulation. (I) The CM-H2DCFDA fluorescence intensity of VNCs of thethird instar larvae of controls (*w1118*) and the indicated genotypes was quantified and normalized to the value of controls (n ≥ 6 animals for each genotype). (J) Representative traces of EJPs evoked by spontaneous motor CPG activity during fictive locomotion obtained from the third instar larvae of controls (*w1118*) and the indicated genotypes. (K–L) Quantification data for burst duration and overall firing time (for bursts of >15 s) per recording minute (n ≥ 7 animals for each genotype). (M) Either neuronal inactivation or relieving oxidative stress does not affect the levels of perisynaptic glutamate of glutamatergic interneurons in *eaat1* mutants. Perisynaptic glutamate was detected by expression of the iGluSnFR reporter driven by *OK371-GAL4*. Averaged iGluSnFR fluorescence intensity was quantified and normalized to the value of controls (n ≥ 4 VNC for each genotype). P values: ns, no significance; *, p<0.05; **, p<0.01; ***, p<0.001; ****, p<0.0001. n: replicate number. Error bars indicate SEM. Statistics: one-way ANOVA with Tukey's post hoc test.
DOI: https://doi.org/10.7554/eLife.47372.026

The following source data and figure supplements are available for figure 6:

**Source data 1.** Source data for *Figure 6*.
DOI: https://doi.org/10.7554/eLife.47372.029

**Figure supplement 1.** Phenotypic characterization of *dsod1* knockdown larvae.
DOI: https://doi.org/10.7554/eLife.47372.027

**Figure supplement 1—source data 1.** Source data for *Figure 6—figure supplement 1*.
DOI: https://doi.org/10.7554/eLife.47372.028

*eaat1* mutants. Consistent with this result from mitotimer, we detected elevated cytosolic ROS levels in *eaat1hypo* mutant muscles, as detected by CM-H2DCFDA staining (*Figure 6C,D*). The ROS increase was normalized after the increased inputs from motor neurons were abrogated by reducing *gluRIID* (*Figure 3* and *Figure 6C,D*) or by expression of *hSOD1* driven by *nSyb-GAL4* or *cha-GAL4* (*Figure 4D–G* and *Figure 6E*). By contrast, expression of *hSOD1* driven by *OK371-GAL4* failed to rescue altered locomotor CPG activity (*Figure 4D–G*) and excess muscular ROS (*Figure 6E*). Therefore, the altered locomotor CPG activity due to loss of *eaat1* can overexcite motor neurons and muscles, promoting oxidative stress in muscles.

To assess whether excess ROS influences muscle strength, we boosted oxidative stress in muscles by knocking down *Drosophila superoxide dismutase 1* (*dsod1*) before measuring muscle contractility and larval locomotion. Muscular expression of *UAS-dsod1-RNAi* using *C57-GAL4* elevated ROS production in muscles but not in other tissues, such as the VNC (*Figure 5F*). The muscle contractility of larvae was measured during spontaneous peristaltic waving using a video-tracking system. Unlike *GAL4* controls, muscles of *dsod1* knockdown larvae could not contract properly (*Figure 6—figure supplement 1A–B*). As a result, these larvae also showed compromised locomotion (*Figure 6G*). A similar locomotion defect was obtained by knocking down *dsod2* (*Figure 6G*), which encodes *Drosophila* mitochondrial Sod. Furthermore, when we expressed *hSOD1* in *eaat1hypo* mutant muscles using *C57-GAL4*, the ROS increase was abrogated (*Figure 6E*) and larval locomotion significantly improved (*Figure 6H*). Note that muscular knockdown of *dsod1* alone did not affect normal synaptic transmission (*Figure 6—figure supplement 1C–E*) or locomotor CPG activity (*Figure 6—figure supplement 1F–I*). Therefore, these results suggest that the increased oxidative stress triggered by tonic motor neuron stimulation upon loss of *eaat1* weakens muscle contraction and larval movement.

However, this significant improvement in larval locomotion of the *eaat1* mutants by muscular expression of *hSOD1* was surprising, as we had expected that aberrant locomotor CPG pattern and impaired muscle contractility together contribute to the compromised locomotion. Previous studies have shown that the contractile status of muscles is coupled with the activation of proprioceptive sensory neurons, through which muscle contraction triggers sensory input to modulate or terminate

locomotor CPG output to motor neurons (*Hughes and Thomas, 2007*; *Kohsaka et al., 2017*; *Song et al., 2007*). Therefore, this scenario raises the attractive possibility that inefficient muscle contraction upon loss of *eaat1* may attenuate sensory feedback inhibition to exacerbate premotor circuit dysfunction. In support of this possibility, muscular expression of *hSOD1* not only relieved the ROS increase in the locomotor CPG circuit (*Figure 6I*), but also reversed the prolonged circuit output (*Figure 6J–L*). We also treated *eaat1* mutants with 2 mM 4-AP to re-activate cholinergic interneurons, which reversed aberrant locomotor CPG activity and defective locomotion (*Figure 5D–E*). Consistent with the effect of muscular expression of hSOD1, the ROS increase in the *eaat1* mutant VNC was significantly relieved by means of 2 mM 4-AP treatment (*Figure 4B*). Interestingly, under this condition, the increase in perisynaptic glutamate in *eaat1* mutants was not affected (*Figure 6M*). A similar result was obtained by treatment with the antioxidant AD4 (*Figure 6M*). Hence, these results indicate that excitotoxicity can induce oxidative stress in the locomotor CPG circuit and muscles and can lead to their dysfunction via a motor circuit-dependent mechanism, and that the feedback arising from inefficient muscle contraction sustains oxidative stress in the locomotor CPG circuit independently of glutamate release.

## The ROS-induced JNK signaling pathway alters NMJ bouton architecture

Growth of *Drosophila* NMJ boutons is linked to the excitation status of motor neurons (*Budnik et al., 1990*; *Oswald et al., 2018a*; *Tsai et al., 2012*). In addition, elevated oxidative stress was previously shown to enhance the formation of NMJ boutons (*Milton et al., 2011*). Therefore, we postulated that, upon Eaat1 depletion, excess CPG stimulation of motor neurons may boost oxidative stress, altering NMJ bouton architecture. As shown in *Figure 7L–M*, *eaat1*<sup>hypo</sup> mutant motor neurons exhibited an increased RFP/GFP ratio of mitotimer compared to that in control motor neurons. This outcome suggests that motor neuron overexcitation does indeed elevate oxidative stress. Furthermore, when excess excitation of *eaat1*<sup>hypo</sup> mutant motor neurons was corrected by feeding larvae with 4-AP (*Figure 5*) or expressing *hSOD1* using *cha-GAL4* or *C57-GAL4* (*Figure 4* and *Figure 6*), the NMJ bouton phenotype was also partially or fully rescued (*Figure 7A–E and K*). Similarly, expression of *hSOD1* in all neurons (using *nSyb-GAL4*), or specifically in glutamatergic neurons (using *OK371-GAL4*) or motor neurons (using *D42-GAL4*), resulted in robust phenotypic rescue (*Figure 7F, H and K*). Moreover, treatment with 15 µg/ml AD4 had the same effect (*Figure 7G and K*). However, expression of *hSOD1* driven by *OK371-GAL4* did not rescue prolonged CPG outputs (*Figure 4*). Hence, our data indicate that the excitotoxicity-induced ROS increase acts cell-autonomously to alter NMJ bouton formation upon loss of *eaat1*.

The conserved c-JUN-N-terminal kinase (JNK) stress signaling pathway is activated by increased oxidative stress to mediate a variety of cellular events, including differentiation, survival, and apoptosis (*Kim and Choi, 2010*). It has also been shown that the ROS-JNK axis positively regulates NMJ bouton growth in *Drosophila* (*Milton et al., 2011*). To investigate whether ROS-dependent activation of JNK signaling drives the NMJ bouton change associated with Eaat1 depletion, we genetically reduced the levels of key components of the JNK signaling pathway in the *eaat1*<sup>hypo</sup> mutant background. Removal of one copy of *bsk* (encoding the *Drosophila* homolog of *JNK*) or *kay* (encoding the *Drosophila* homolog of *c-FOS*) from *eaat1*<sup>hypo</sup> mutants resulted in the restoration of normal NMJ bouton morphology (*Figure 7I and K*). Consistent with these results, motor-neuron-specific expression of the dominant-negative form of c-Fos (c-Fos<sup>DN</sup>) rescued the bouton phenotype (*Figure 7J–K*). By contrast, reducing *jra* that encodes the *Drosophila* homolog of *c-JUN* had no effect (*Figure 7K*), which is consistent with a previous finding (*Milton et al., 2011*). Restoration of NMJ bouton morphology could further normalize the release probability of *eaat1*<sup>hypo</sup> mutant NMJs (*Figure 7N*), but it did not have an impact on CPG circuit dysregulation (*Figure 7O*) or compromised locomotion (*Figure 7P*). Hence, these findings indicate that ROS-activated JNK signaling cell-autonomously causes abnormal NMJ bouton formation in *eaat1* mutants, and they support premotor circuit dysfunction as being the primary cause of NMJ bouton abnormalities and compromised locomotion upon loss of *eaat1*.

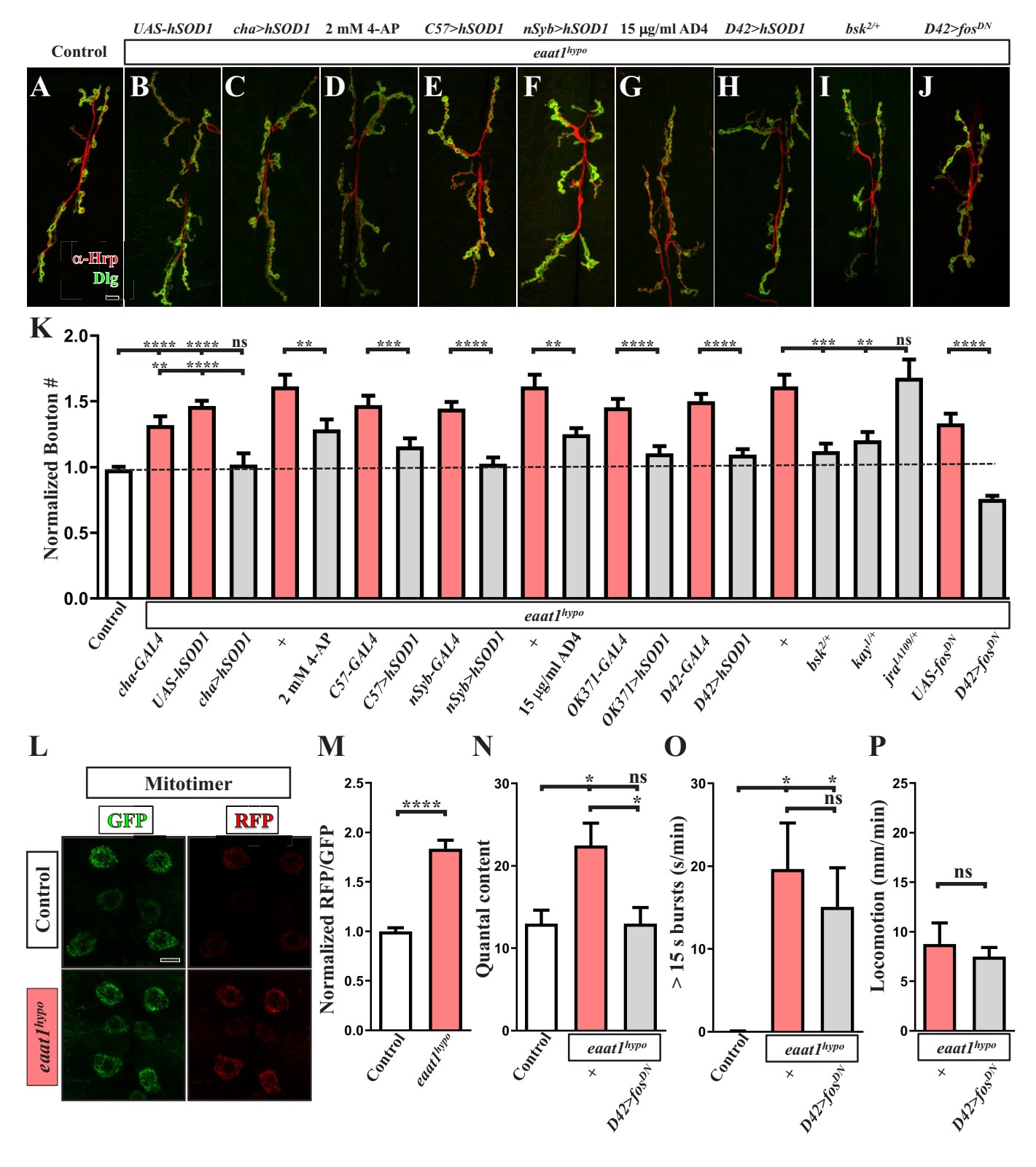

**Figure 7.** Premotor circuit dysfunction causes altered NMJ bouton architecture via ROS-dependent activation of the JNK signaling pathway upon loss of *eaat1*. (A–J) Representative confocal images of NMJs co-stained with α-HRP (red) and α-Dlg (green) obtained from third instar larvae of controls (*w1118*) and the indicated genotypes. Scale bar: 10 μm. (K) Quantification data for the number of NMJ boutons per muscle area normalized to the value of controls (n ≥ 7 NMJs of A2 muscles 6 and 7 derived from n ≥ 7 animals for each genotype). (L–M) Loss of *eaat1* increases mitochondrial ROS in

*Figure 7 continued on next page*

*Figure 7 continued*

motor neurons. (L) Representative confocal images of motor neurons of third instar larvae expressing mitotimer using *D42-GAL4* obtained from controls (*w^1118^*) and *eaat1^hypo/hypo^* mutants. Scale bar: 5 μm. (M) The RFP/GFP ratio of mitotimer was quantified and normalized to the value of controls (n ≥ 10 animals for each genotype). (N,O) Expression of *fos^DN^* in motor neurons rescues the NMJ bouton phenotype based on normalized quantal content, but does not affect premotor circuit dysregulation in *eaat1^hypo^* mutants. (N) Quantification data for quantal content recorded from A3 muscle 6 of third instar larvae of controls (*w^1118^*) and the indicated genotypes with 0.2 Hz electric stimulation in 0.5 mM Ca$^{2+}$-containing HL3 solution (n ≥ 6 animals for each genotype). (O) Quantification data for overall firing time (for bursts of >15 s) per recording minute (n ≥ 8 animals for each genotype). (P) Quantification data for larval locomotion of *eaat1^hypo^* mutants and *eaat1^hypo^* mutants who express *fos^DN^* in motor neurons (n ≥ 20 animals for each genotype). P values: ns, not significant; *, p<0.05; **, p<0.01; ***, p<0.001; ****, p<0.0001. n: replicate number. Error bars indicate SEM. Statistics: Student's *t*-test or one-way ANOVA with Tukey's post hoc test.

DOI: https://doi.org/10.7554/eLife.47372.030

The following source data is available for figure 7:

**Source data 1.** Source data for *Figure 7*.
DOI: https://doi.org/10.7554/eLife.47372.031

## Discussion

In this work, we utilized a fly model of glutamate excitotoxicity induced by loss of *Drosophila eaat1* to explore the impact of glutamate excitotoxicity on the integrity of the motor system. We report a circuit-dependent feedback mechanism for increasing ROS that mediates excitotoxicity to alter pre-motor circuit activity, NMJ architecture, and motor function. As summarized in *Figure 8*, glutamate excitotoxicity initially alters locomotor CPG activity and hence prolongs CPG output bursts onto motor neurons by ROS-mediated inactivation of the cholinergic interneurons constituting the CPG circuit. Then, tonic premotor stimulation triggers activity-dependent ROS overproduction in both

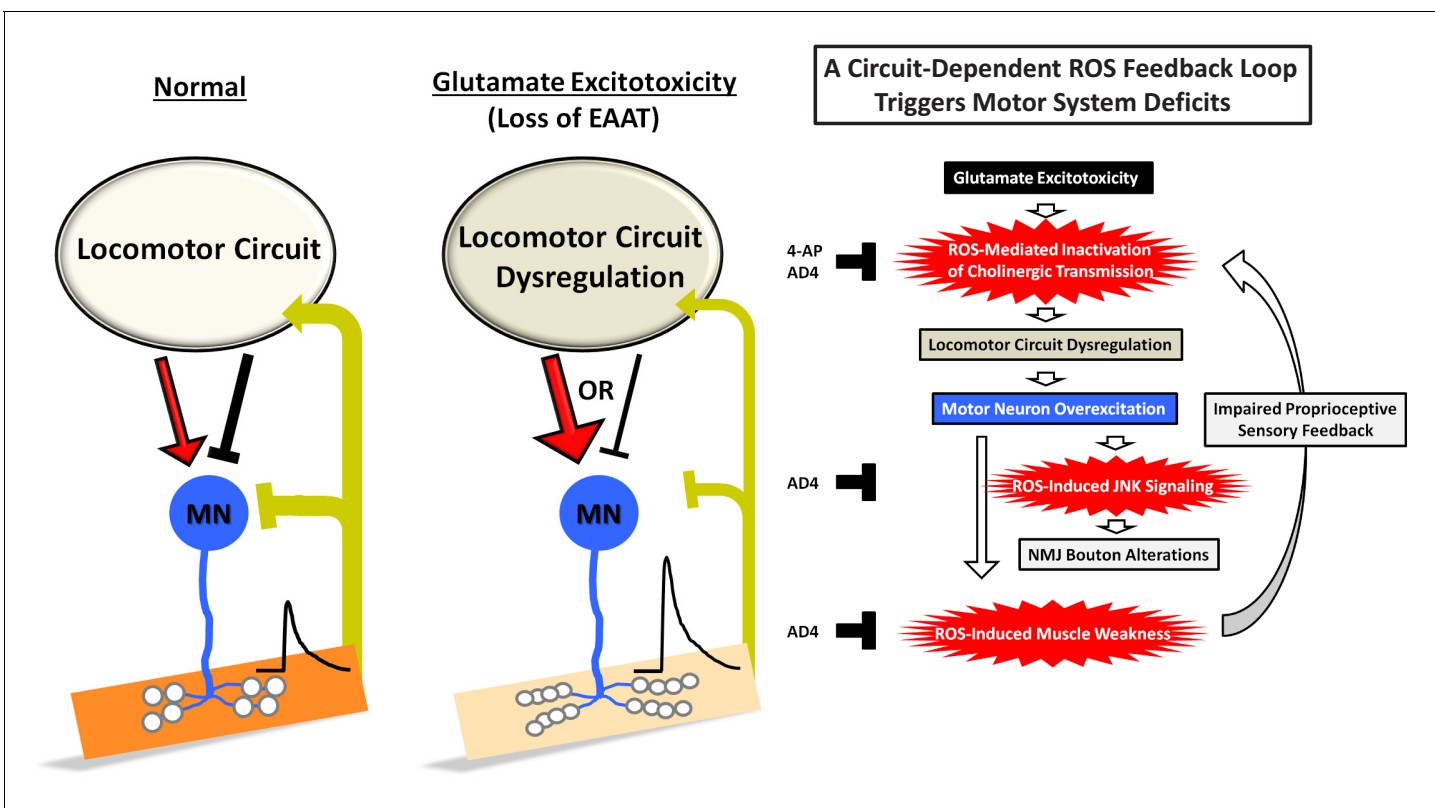

**Figure 8.** A schematic model of a circuit-dependent ROS feedback loop under glutamate excitotoxicity induced by loss of *eaat1*. The details are presented in the discussion section.
DOI: https://doi.org/10.7554/eLife.47372.032

motor neurons and muscles. In muscles, the increased ROS level gradually dampens muscle contractility and consequent sensory input back to the locomotor CPG circuit, with this feedback strengthening ROS accumulation within the CPG circuit to exacerbate circuit dysfunction. Thus, a positive feedback loop between ROS production in the CPG circuit and muscles is established. Finally, in motor neurons, the induced ROS activate JNK stress signaling to promote abnormal NMJ bouton outgrowth and strength. Apart from genetic rescue, pharmacological treatment with the antioxidant AD4 or the $K^+$ channel blocker 4-AP can also significantly alleviate these motor-system deficits.

## Glutamatergic transmission regulates the *Drosophila* locomotor CPG circuit

The locomotor CPG circuit for *Drosophila* larval feed-forward locomotion is positioned in the VNC and is activated by input from the central brain (*Cattaert and Birman, 2001*). Furthermore, acute treatment of dissected *Drosophila* larvae with non-competitive NMDA antagonists has been shown to reduce the initial output burst duration of the locomotor CPG and eventually abolishes all output activity (*Cattaert and Birman, 2001*), suggesting that glutamatergic transmission drives locomotor CPG activity and positively controls its output burst duration during larval movement. Consistent with these latter results, we have shown herein that VNC-restricted expression of *eaat1-venus* using *tsh-GAL4* could reverse the prolonged CPG output burst (*Figure 3*) but not the reduced CPG output frequency (*Figure 1—figure supplement 3*) in *eaat1* mutants, indicating that central brain removal of *eaat1* reduced burst frequency, whereas VNC removal markedly extended burst duration. The exact neuronal identity and network connections that build up the core components of the *Drosophila* larval locomotor CPG circuit remain unknown (*Kohsaka et al., 2017*; *Ohyama et al., 2015*). Intriguingly, the phenotype of prolonged CPG output has also been reported under conditions in which the motor neuron inputs from proprioceptive sensory neurons or period-positive median segmental interneurons (PMSI) are limited (*Kohsaka et al., 2014*; *MacNamee et al., 2016*; *Song et al., 2007*). Moreover, RNAi-mediated knockdown of *eaat1* extends PMSI-evoked inhibitory postsynaptic currents in motor neurons (*MacNamee et al., 2016*). We also noticed an increase in extrasynaptic glutamate at the axonal synapses of PMSI when *eaat1* is lost (unpublished data), raising an alternative possibility that excess extrasynaptic glutamate may desensitize GluClα to further diminish sensory inhibition feedback. However, we found that reducing *gluRIID* but not *gluClα* in the *eaat1* mutant background shortened prolonged CPG output (*Figure 3* and *Figure 3—figure supplement 1*). Hence, upon loss of *eaat1*, glutamate-mediated excitotoxicity mainly contributes to locomotor CPG circuit dysfunction. In this regard, the CPG outputs to motor neurons may be elongated and/or the potential inhibition from CPG output to PMSI or its upstream interneurons may be abrogated (*Figure 8*). Further experiments will be needed to unravel the detailed mechanism operating in the CPG circuit upon loss of *eaat1*.

## ROS-mediated inactivation of cholinergic transmission underlies the locomotor CPG dysregulation in *eaat1* mutants

Targeted relief of the increased ROS in cholinergic interneurons by genetic approaches could significantly alleviate altered CPG activity arising from either *eaat1* mutation (*Figure 4*) or short-term exposure to $H_2O_2$ (*Figure 5*), indicating that a subset of cholinergic interneurons, which presumably constitutes the locomotor CPG circuit, is vulnerable to and influenced by the ROS increase. Our temporal rescue experiments further suggest that the effect of the ROS increase on circuit activity is acute and reversible (*Figure 5*). In support of the fact that ROS is known to reduce the inactivation of voltage-gated potassium channels in neurons (*Sahoo et al., 2014*; *Sesti et al., 2010*), long-term food-mediated feeding of (or even short-term exposure to) the $K^+$ channel blocker 4-AP led to a restoration of CPG activity in *eaat1* mutants (*Figure 4* and *Figure 5*). Thus, temporal hypoexcitability of cholinergic interneurons most probably underlies ROS-induced locomotor CPG dysfunction upon *eaat1* loss. Interestingly, immediate blockade of glutamatergic transmission shortens the burst duration of the CPG output (*Cattaert and Birman, 2001*). Under this scenario, it is expected that shortened rather than prolonged CPG burst durations should occur upon loss of *eaat1*. Thus, it is likely that the induced ROS may occur in a restricted way in a certain subset of cholinergic interneurons, resulting in uneven suppression of cholinergic transmission in the locomotor CPG circuit.

The GABA neurotransmitter has a crucial role in neuronal inhibition in the central nervous system through its actions on GABA receptors (*Farrant and Nusser, 2005*). Notably, regulation of GABAergic transmission by redox signaling is increasingly recognized (*Beltrán González et al., 2019*). ROS, especially those derived from mitochondrial respiration, act to strengthen the neuronal inhibition mediated by GABA$_A$ receptors (*Accardi et al., 2014*; *Penna et al., 2014*). Thus, we do not exclude an alternative possibility that, if those ROS-vulnerable cholinergic interneurons also receive GABAergic input, the increased ROS may silence cholinergic transmission of the locomotor CPG circuit by strengthening GABA-mediated inhibition.

## ROS-induced muscle weakness initiates a circuit-dependent ROS feedback loop

The pathological roles of ROS in the regulation of skeletal muscles have been studied extensively (*Barbieri and Sestili, 2012*; *Moylan and Reid, 2007*; *Powers et al., 2011*), and most targets of redox signaling in skeletal muscle participate in muscle contraction. For instance, excess ROS can modulate SR calcium ATPase (SERCA) and ryanodine receptor (RyR) activity (*Gutiérrez-Martín et al., 2004*; *Xu et al., 1997*; *Zima and Blatter, 2006*), both of which control the Ca$^{2+}$ homeostasis of sarcoplasmic reticulum. ROS exposure can also oxidize some myofilament proteins, such as myosin heavy chain and troponin C and, in turn, can impair their functions (*Coirault et al., 2007*; *Plant et al., 2000*; *Yamada et al., 2006*). Consistent with these findings, the ROS increase dampens the muscle contractility of *Drosophila* larvae. We found that mitochondrial and cytosolic ROS levels increase upon excess motor neuron stimulation when *eaat1* is lost (*Figure 6*). Moreover, while increasing ROS by *dsod1* knockdown reduced muscle contractility and movement velocity, relieving excess ROS in *eaat1* mutant muscles improved locomotion (*Figure 6*). In the motor system, muscles are not only recognized as the end executive tissues for body movement, but also have a crucial role in triggering proprioceptive sensory feedback input to the central circuit (*Hughes and Thomas, 2007*; *Kohsaka et al., 2017*; *Song et al., 2007*). Recent studies in *Drosophila* have also revealed that proprioceptive sensory feedback plays a vital role in tuning locomotor circuit activity in the homeostatic adjustment of *Drosophila* larval crawling (*Oswald et al., 2018a*) and in a *Drosophila* model of amyotrophic lateral sclerosis (ALS) (*Held et al., 2019*). Unexpectedly, we found that, upon glutamate excitotoxicity, ROS-induced muscle weakness can cause inefficient sensory feedback input to worsen the ROS burden and can negatively impact the functioning of the central locomotor network. Therefore, under pathological conditions, impaired muscle activity can serve as a key mediator for initiating the ROS feedback loop between the CPG circuit and muscles, which may contribute to network dysfunction in excitotoxicity-associated diseases.

## Excitotoxicity-induced premotor circuit dysfunction elicits activity-dependent synaptic changes

ROS are known to activate the JNK/AP-1 signaling pathway (*Zhang et al., 2016*) that regulates synaptic formation and strength in *Drosophila* (*Collins et al., 2006*; *Milton et al., 2011*; *Sanyal et al., 2003*; *Sanyal et al., 2002*; *Shen and Ganetzky, 2009*; *West et al., 2015*). The mutation in *Drosophila spinster* (*spin*), which encodes a late endosome and lysosome protein (*Dermaut et al., 2005*; *Sweeney and Davis, 2002*), causes impaired lysosomal activity and a consequent ROS burden, leading to synaptic bouton outgrowth by activating the JNK signaling pathway (*Milton et al., 2011*). Intriguingly, c-FOS but not c-JUN is important for bouton outgrowth under *spin* loss (*Milton et al., 2011*). Similarly, we found that the synaptic bouton phenotypes of *eaat1* mutants are dependent on ROS and c-FOS activity (*Figure 7*). Interestingly, *Milton et al., 2011*) have shown that 'constitutive' boosting of mitochondria-derived ROS under loss of *dsod2* or after paraquat treatment can also promote bouton growth, but in that case it requires both c-FOS and c-JUN activities. By contrast, in *eaat1* mutants, the altered CPG pattern possibly elicits a pulsed increase of mitochondrial ROS. Thus, it may be postulated that different resources and temporal generation of ROS may be responsible for engaging different cellular signaling processes. In support of this notion, in addition to mitochondria, NADPH oxidases provide another major source of ROS to control diverse cellular processes (*Brennan et al., 2009*; *Oswald et al., 2018b*; *Zhang et al., 2016*). Recently, DJ-1β, a Parkinson's disease-linked protein, has been identified as a redox sensor that mediates the

mitochondrial ROS regulating activity-dependent synaptic plasticity at *Drosophila* NMJ (*Oswald et al., 2018a*). It will be interesting to further investigate the underlying mechanisms in detail.

## Potential relevance of ROS-induced motor-circuit dysregulation for neurodegenerative diseases

Downregulation of EAAT2 has been demonstrated in patients with Alzheimer's disease (*Jacob et al., 2007*; *Li et al., 1997*) or amyotrophic lateral sclerosis (ALS) (*Rothstein et al., 1995*), as well as in ALS rodent models (*Bendotti et al., 2001*; *Bruijn et al., 1997*; *Howland et al., 2002*; *Tong et al., 2013*). ALS is a fatal adult-onset disease that predominantly causes NMJ denervation, motor neuron degeneration, and compromised motor function (*Taylor et al., 2016*). Spinal removal of mouse EAAT2 is sufficient to elicit motor neuron death (*Sugiyama et al., 2017*; *Sugiyama and Tanaka, 2018*). Transgenic expression of EAAT2 (*Guo et al., 2003*) or treatment with the small compound LDN/OSU-0212320 (*Kong et al., 2014*), which mainly increases translation of EAAT2 mRNA, improves the motor performance of an ALS mouse model expressing hSOD1G93A. However, a recent clinical study testing ceftriaxone, an FDA-approved β-lactam antibiotic that can transcriptionally promote *EAAT2* expression (*Lee et al., 2008*; *Rothstein et al., 2005*), in ALS patients concluded that this drug treatment had no therapeutic effect (*Cudkowicz et al., 2014*). Therefore, it is questionable whether increasing EAAT2 expression represents a feasible therapeutic strategy for ALS. It has been argued, however, that EAAT2 downregulation largely occurs at the posttranslational and not at the transcriptional level in ALS (*Bristol and Rothstein, 1996*; *Kong et al., 2014*). There was no evidence for increased EAAT2 in patients treated with ceftriaxone (*Cudkowicz et al., 2014*), and ceftriaxone treatment only slightly increases protein levels of EAAT2 in hSOD1G93A mice (*Kong et al., 2014*). In addition, the efficacy of ceftriaxone in hSOD1G93A mice is not consistent among different studies (*Kong et al., 2012*; *Kong et al., 2014*; *Rothstein et al., 2005*; *Scott et al., 2008*). Hence, more investigations will be needed to strengthen evidence for the pathogenic contribution of EAAT2 dysfunction in ALS.

Oxidative stress is known as a hallmark of Alzheimer's disease (*Wang et al., 2014*; *Zhao and Zhao, 2013*), Parkinson's disease (*Blesa et al., 2015*), and ALS (*Barber and Shaw, 2010*). During aging, neurons are thought to be susceptible to excitotoxicity (*Lewerenz and Maher, 2015*), and the nervous system and muscles are vulnerable to ROS accumulation because of high oxygen consumption demand (*Jackson and McArdle, 2011*; *Liguori et al., 2018*). Administration of antioxidants can improve the motor function of hSOD1G93A mice (*Andreassen et al., 2000*; *Aoki et al., 2011*; *Crow et al., 2005*; *Gurney et al., 1996*; *Ito et al., 2008*; *Matthews et al., 1998*) and ALS patients (*Abe et al., 2014*; *The Writing Group, 2017*). Nonetheless, how oxidative stress is produced in ALS and how this burden is involved in disease pathogenesis is not well understood. Interestingly, in our study, *Drosophila* Eaat1 depletion was shown to cause ALS-like characteristics, including motor neuron excitotoxicity, NMJ bouton abnormalities, muscle weakness, and compromised motor performance. In the future, it will be worth exploring whether ROS-induced motor circuit dysfunction might also participate in ALS progression and age-dependent motor system decline.

It has previously been shown that reduced excitability of proprioceptive sensory neurons and cholinergic interneurons is causative of locomotor CPG circuit dysfunction and compromised locomotion in *Drosophila smn* mutants, which are used as a *Drosophila* model of spinal muscular atrophy (SMA), a motor neuron disease of juveniles (*Imlach et al., 2012*; *Lotti et al., 2012*). Increasing neuronal excitability by 4-AP treatment reverses these motor system defects (*Imlach et al., 2012*; *Lotti et al., 2012*). Intriguingly, our data show that long-term food-mediated feeding of (or even short-term exposure to) the K$^+$ channel blocker 4-AP also rescued altered locomotor CPG activity in *eaat1* mutants (*Figure 5*). Notably, although the precise mechanisms are unknown, 4-AP has been used to treat several motor system-related disorders such as spinal cord injury (*Hayes, 2007*), Lambert-Eaton syndrome (*Quartel et al., 2010*), and hereditary canine spinal muscular atrophy (*Pinter et al., 1997*), and it is an FDA-approved therapy for multiple sclerosis (*Chwieduk and Keating, 2010*; *Hayes, 2007*). Thus, as supported by our findings, it seems plausible that neuronal hypoexcitability may be a shared mechanism underlying the motor-system defects displayed in motor-related disorders.

## Materials and methods

### Fly strains

Flies were reared in regular food at 25°C. To obtain third instar *eaat1*$^{hypo}$ mutants, eggs were laid on grape juice plates covered with yeast paste, and larvae were grown on a fresh plate with yeast paste at 25°C. The EMS-mutagenized mutants were kindly provided by Hugo J Bellen. Other fly stocks were obtained from the Bloomington *Drosophila* Stock Center (https://bdsc.indiana.edu/) and the Vienna *Drosophila* RNAi Center (https://stockcenter.vdrc.at/control/main): *nSyb-GAL4* (*Pauli et al., 2008*); *OK371-GAL4* (*Mahr and Aberle, 2006*); *D42-GAL4* (*Yeh et al., 1995*); *cha-GAL4* (*Salvaterra and Kitamoto, 2001*); *gad1-GAL4* (*Ng et al., 2002*); *repo-GAL4* (*Sepp et al., 2001*); *alrm-GAL4* (*Doherty et al., 2009*); *NP2222-GAL4* (*Sugimura et al., 2003*); *tsh-GAL4* (*Tomoyasu et al., 1998*); *C57-GAL4* (*Packard et al., 2002*); *GMR49G06-GAL4* (Janelia GAL4) (*Pfeiffer et al., 2008*); *tub-Gal80*$^{ts}$ (*McGuire, 2003*); *repo-LexA* (*Lai and Lee, 2006*); *UAS-CD4::spGFP*$^{1-10}$ and *LexAOP-CD4::spGFP*$^{11}$ (*Gordon and Scott, 2009*); *UAS-hSOD1* (*Watson et al., 2008*); *UAS-fos*$^{DN}$ and *UAS-Jun*$^{DN}$ (*Eresh et al., 1997*); *UAS-mitoTimer* (*Laker et al., 2014*); *UAS-vglut-RNAi* (Vienna *Drosophila* RNAi Center, #v2574); *UAS-dsod1-RNAi* (Vienna *Drosophila* RNAi Center, #v31551); *UAS-dsod2-RNAi* (Bloomington *Drosophila* Stock Center, #32496); *gluRIID*$^{e01443}$ (*Featherstone, 2005*); *kay*$^1$ and *kay*$^2$ (*Riesgo-Escovar and Hafen, 1997*); *jra*$^{IA109}$ (*Hou et al., 1997*); *bsk*$^2$ (*Sluss et al., 1996*); *UAS-eaat1-venus* (*Parinejad et al., 2016*); *eaat1*$^{SM2}$ (*Stacey et al., 2010*); *UAS-iGluSnFR* (*Stork et al., 2014*); and *gluClα*$^{glc1}$ (*Kane et al., 2000*).

### DNA cloning and genomic sequencing

The genomic DNA of third instar larvae of homozygous *eaat1*$^{hypo}$ mutants was extracted using a Gentra Puregene Tissue Kit (Qiagen). The primers were designed to cover 500–600 base pairs (bp) of the *eaat1* locus. PCR-amplified products were sequenced. No mutation was found in exons of the *eaat1* gene in the respective *eaat1*$^{hypo}$ chromosome. A 428-bp insertion of the long transcribed region (LTR) of the *roo* transposon was identified in intron 10 of the *eaat1* gene. The sequence of *human EAAT1-GFP* was PCR-amplified from pGFAP-CITE-hEAAT2-EGFP plasmid, which was kindly provided by Dr. Chien-Liang Glenn Lin, using primers (forward primer GGGGTACCA CCATGGCA TCTACGGAAGGTGCC; reverse primer TTTCTAGATTACTTGTACA GCTCGTCC). The PCR fragment was subcloned into the *Kpn*I and *Xba*I sites of the *pUAST* vector. This construct was microinjected into early embryos according to a standard transgenesis protocol.

### Immunohistochemistry and Western blotting

For α-Eaat1 immunostaining, we fixed the samples with Bouin's fixative for 2 min. For other immunostaining, the samples were fixed with 4% paraformaldehyde for 20 min. Primary antibodies were used as follows: mouse α-Dlg (Developmental Studies Hybridoma Bank 4F3, 1:100); mouse α-Bruchpilot (Developmental Studies Hybridoma Bank nc82, 1:100); mouse α-dCSP2 (Developmental Studies Hybridoma Bank 6D6, 1:100); mouse α-Repo (Developmental Studies Hybridoma Bank 8D12, 1:100); rat α-Elav (Developmental Studies Hybridoma Bank 7E8A10, 1:400); chicken α-GFP (Abcam, 1:1000); rabbit α-RFP (Clontech, 1:200); rat α-RFP (Chromotek, 1:1000) and rabbit α-HRP conjugated with Alexa Fluor 488, Cy3 or Cy5 (Jackson ImmunoResearch Laboratories, 1:300). Secondary antibodies conjugated to Alexa Fluor 488, Alexa Fluor 555, and Alexa Fluor 647 (Invitrogen and Jackson ImmunoResearch) were used at 1:500. Images were captured with a Zeiss 780 confocal microscope and processed using LSM Zen and Image J software (National Institutes of Health). For Western blotting, we suspended ten L3 brains in 50 µl ice-cold 1xSDS sample buffer, followed by homogenization and boiling for 5 min. Five brain lysates were loaded into a 15% SDS-PAGE gel. We used rabbit α-Eaat1 (1:20,000) (*Peco et al., 2016*) and mouse α-tubulin (Sigma, 1:10,000) as primary antibodies. HRP-conjugated secondary antibodies (Jackson ImmunoResearch Laboratories) were used at 1:10,000.

### Live imaging

Third instar larvae were dissected in zero calcium HL3 buffer (70 mM NaCl, 5 mM KCl, 10 mM MgCl$_2$, 10 mM NaHCO$_3$, 5 mM HEPES, 115 mM sucrose, 5 mM Trehalose, pH 7.2) at room temperature. Imaging was performed in zero calcium HL3 buffer and captured by a 60X water-immersion

objective and EMCCD camera (iXon, Andor) mounted on a SliceScope Pro 6000 (Scientifica) microscope using MetaFlour software (Molecular Devices).

## CM-H2DCFDA staining

Third instar larvae were dissected in zero calcium HL3 buffer. CM-H2DCFDA was detected as previously described (*Nguyen et al., 2018*). In brief, 1x PBS buffer containing 10 µM CM-H2DCFDA (ThermoFisher) and 2 mM $Ca^{2+}$ was added to larval fillets, followed by orbital shaking for 15 min in the dark. Excess dyes were removed by several washes of zero calcium HL3 buffer. Subsequently, the samples were fixed with 60% methanol in 1 x PBS buffer for 10 min. The images were captured by Zeiss 780 confocal microscopy and analyzed using Image J.

## Electrophysiology

Evoked excitatory junctional potential (EJP) was recorded as previously described (*Yao et al., 2009*). In brief, third instar larvae were dissected in zero calcium HL3 buffer at room temperature, followed by incubation in 0.5 mM $Ca^{2+}$ HL3 solution for 5–10 min prior to recording. The mean resistance value for the recording electrode was ~40 MΩ when 3M KCl solution was used as the electrode solution. All recordings were obtained from A3 muscle 6. Resting membrane potentials of muscles were held at less than −60 mV. EJPs were amplified with an Axoclamp 900A amplifier (Axon Instruments, Foster City, CA) under bridge mode and filtered at 10 kHz. EJPs were analyzed using the pClamp 10.6 software (Axon Instruments). Averaged EJP amplitude was calculated from the amplitudes of 80 EJPs in one consecutive recording. Miniature EJP recordings were performed in HL3 solution containing 0.5 mM $Ca^{2+}$ and 5 µM tetradotoxin (TTX), and analyzed using pClamp 10.6 software. Spontaneous motor central pattern generator (CPG) activity was measured as previously described (*Imlach et al., 2012*; *McKiernan, 2013*). Third instar larvae were dissected in zero calcium modified HL3 solution (70 mM NaCl, 5 mM KCl, 4 mM $MgCl_2$, 10 mM $NaHCO_3$, 5 mM HEPES, 5 mM Trehalose, and 115 mM sucrose, pH 7.2). Subsequently, EJPs were recorded from A3 muscle 6 in 1 mM $Ca^{2+}$-containing HL3 solution for 10 min with a long sharp electrode. EJPs were analyzed using the pClamp 10.6 software. A successful burst was defined as a burst containing ≥15 consecutive EJPs with <1 s intervals between them. For $H_2O_2$ exposure, larval fillets were bathed in the HL3 solution containing 1 mM $Ca^{2+}$ and 0.006% $H_2O_2$ for ~3 min after dissection, and recordings were done in the same solution for 10 min. For 4-AP treatment, we preincubated dissected larval fillets in HL3 solution containing 1 mM $Ca^{2+}$ and 0.2 mM 4-AP, and recordings were conducted in new HL3 solution containing 0.2 mM 4-AP or/and 0.006% $H_2O_2$ for 10 min.

## Larval locomotion

We adopted a previously described protocol for our locomotion assay (*Aleman-Meza et al., 2015*). In brief, third instar larvae were placed on a 6 mm Petri dish containing 1.0% agar for 10 min, during which time they were habituated and starved. One to five larvae were then transferred onto a new plate, and larval locomotion was traced for 30 s using a camera mounted on a SteREO DiscoveryV8 (Zeiss) stereomicroscope using AxioVision software (Zeiss). For data quantification, locomotion tracks were superimposed and analyzed using Image J.

## Drug treatment

Embryos of wild-type controls and *eaat1* mutants were placed on a grape juice plate topped with yeast paste containing 15 µg/ml AD4 (Sigma) or 2 mM 4-AP (Sigma). Hatched larvae were grown at 25°C on plates changed daily with the same concentrations of compounds throughout the larval stage.

## Statistics

Paired and multiple datasets were compared by Student's *t*-test and one-way ANOVA, respectively. All data analyses were conducted using GraphPad Prism 8.0.

## Acknowledgements

We are grateful to Hugo Bellen for sharing EMS-mutagenized fly stocks. We thank DJ van Meyel for sharing UAS-eaat1-venus flies and Eaat1 antibody. We appreciate the critical readings of Henry Sun, Ruey-Hwa Chen, Yijuang Chern, and Guang-Chao Chen. We thank Brian McCabe, Aaron Diantonio, Jimena Berni, Marc Freeman, Tzu-Yang Lin, Bloomington Stock Center, Vienna *Drosophila* RNAi Center, and the Developmental Studies Hybridoma Bank for reagents. We thank Tzu-Li Yen and Shiu-Hui Lin for assisting with the NMJ bouton screen; Chien-Liang Glenn Lin for providing human EAAT2-egfp cDNA; Hung-Lune Chen for assisting with DNA construction; Chiou-Yang Tang for aiding us with microinjections, and the Taiwan FlyCore for assistance. This work was supported by grants from the Ministry of Science and Technology (101–2311-B-001–015-MY3, 105–2311-B-001–076, 107–2311-B-001–003-MY3, 106-0210-01-15-02, and 107-0210-01-19-01) and Academia Sinica (AS-103-TP-B05).

## Additional information

### Funding

| Funder | Grant reference number | Author |
| --- | --- | --- |
| Academia Sinica | AS-103-TP-B05 | Chi-Kuang Yao |
| Ministry of Science and Technology, Taiwan | 101-2311-B-001-015-MY3 | Chi-Kuang Yao |
| Ministry of Science and Technology, Taiwan | 105-2311-B-001-076 | Chi-Kuang Yao |
| Ministry of Science and Technology, Taiwan | 107-2311-B-001-003-MY3 | Chi-Kuang Yao |
| Ministry of Science and Technology, Taiwan | 106-0210-01-15-02 | Chi-Kuang Yao |
| Ministry of Science and Technology, Taiwan | 107-0210-01-19-01 | Chi-Kuang Yao |

The funders had no role in study design, data collection and interpretation, or the decision to submit the work for publication.

### Author contributions

Jhan-Jie Peng, Conceptualization, Resources, Data curation, Software, Formal analysis, Validation, Investigation, Visualization, Methodology, Project administration; Shih-Han Lin, Conceptualization, Resources, Data curation, Software, Formal analysis, Supervision, Validation, Investigation, Visualization, Methodology, Project administration; Yu-Tzu Liu, Hsin-Chieh Lin, Tsai-Ning Li, Formal analysis, Methodology; Chi-Kuang Yao, Conceptualization, Resources, Data curation, Software, Formal analysis, Supervision, Funding acquisition, Validation, Investigation, Visualization, Methodology, Writing—original draft, Project administration, Writing—review and editing

### Author ORCIDs

Chi-Kuang Yao  https://orcid.org/0000-0003-0977-4347

### Decision letter and Author response

Decision letter https://doi.org/10.7554/eLife.47372.035
Author response https://doi.org/10.7554/eLife.47372.036

## Additional files

### Supplementary files

• Transparent reporting form
DOI: https://doi.org/10.7554/eLife.47372.033

## Data availability

Source data files for all figures and figure supplements have been uploaded.

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
