## [Decision Letter]

Thank you for submitting your article "Circuit-dependent ROS propagation mediates glutamate excitotoxicity to elicit motor system deficits in *Drosophila*" for consideration by *eLife*. Your article has been reviewed by three peer reviewers, one of whom is a member of our Board of Reviewing Editors, and the evaluation has been overseen by Ronald Calabrese as the Senior Editor. The reviewers have opted to remain anonymous.

The reviewers have discussed the reviews with one another and the Reviewing Editor has drafted this decision to help you prepare a revised submission.

Summary:

In this paper, the authors characterize a hypomorphic mutant of the *Drosophila*excitatory amino acid transporter (*eaat1*). The loss of function of the human Homolog EAAT2 has been implicated in the pathogenesis of the motor neuron disease, ALS. The authors show that loss of *eaat1* in glia of the VNC of larvae result in increased glutamate within the VNC and the specifically the locomotor CPG circuit of larvae. This leads to increased ROS in Cholinergic interneurons resulting in electophysiological defects within the CPG and locomotion defects. The motor neurons within this circuit suffer from hyperexcitability and also show morphological deficits. The authors propose a model whereby ROS production in different parts of the CGP circuit and locomotor system (cholinergic neurons or muscle) can propagate in a feed forward fashion and cause overall disruption of the motor pattern generating circuit.

Essential revisions:

1) There was serious concern from the three reviewers about the connections to ALS and they did not acknowledge the relevance for human ALS. The analysis of *eaat1* and the findings of how ROS are sculpting the circuit were, on the other hand, evaluated as very exciting:

a) The exact relevance of the decrease in a glutamate transporter in the context of ALS is unclear. Despite the fact that lower levels of EAAT2 have been reported in ALS, treatment with a compound that increases the expression of the EAAT2 transporter (ceftriaxone) had no therapeutic effect in ALS patients (Cudkowicz et al., 2014; not cited in this manuscript). No mutations have been found in EAAT2 and the down-regulation during the disease and in ALS models could be a consequence of the disease process, rather than a major cause of the selective motor neuron death.

b) It is far from clear whether the phenotype observed in the *Drosophila* larvae has anything to do with ALS. The most important hallmark of ALS, selective motor neuron death, is not observed. It is not clear whether changes in the number and size of synaptic boutons, increases in the evoked excitatory junctional potentials and overexcitation of the motor neurons have anything to do with what happens during ALS. Only the motor deficits seem to be in line with this disease but it needs to be proven whether the cause of these motor defect is in any way related to what happens in ALS patients.

c) Suggesting that antioxidants could be used as a therapeutic strategy for ALS is not very original and has been suggested many times before. Several antioxidants have been extensively tested in ALS without any success. The potassium channel blocker (4-AP) was also suggested before as a therapeutic strategy. However, this was mainly based on the hypoexcitability phenotype observed in motor neurons differentiated from iPSCs obtained from familial ALS patients. This defect was cell autonomous and had no direct relation with excitotoxicity and/or the interaction with other cell types.

It is critical to reframe the revised manuscript without the strenuous connection to ALS. This will require major rewriting, but we am confident this is feasible within the revision period.

Other important points:

2) Can the authors use an independent *eaat1* mutant or a VNC glia-specific knockdown of *eaat1* and test it in some of the assays to show independent verification of the phenotype. In addition, Figure 2D needs a neuronal control.

3) Discussion, fourth paragraph: The authors write that they find impaired muscle contraction caused by excess ROS exacerbates premotor circuit dysregulation. They showed only muscle contraction weakness in animals with reduced dSOD in the muscle. However, no data for premotor circuit dysregulation or increased ROS generation is shown. Please include also other phenotypes (burst activity, quantal content, ROS sensor) for this mutant condition.

4) The authors claim that ROS generation in cholinergic neurons leads to attenuation of K^+^ channel inactivation, resulting in hypoexcitability. Blocking potassium channels by 4-AP rescues burst phenotypes but also rescues the increased ROS phenotype (Figure 4B). How can the blocking of K^+^ channels result in reduction of ROS caused by glutamate excitotoxicity? Is this via the feedback loop? This could be tested by specifically reducing dSOD in the muscle and treating these animals with 4-AP. This should not rescue ROS in the CPG.

5) Neuronally expressing hSOD1 can fully suppress muscle ROS and muscle-expression of hSOD1 can fully suppress neuronal ROS. Can the author explain this?

6) Several reviewers also found that the quality of presentation of the data could be improved.

---

## [Author Response]

Essential revisions:1) There was serious concern from the three reviewers about the connections to ALS and they did not acknowledge the relevance for human ALS. The analysis of eaat1 and the findings of how ROS are sculpting the circuit were, on the other hand, evaluated as very exciting:a) The exact relevance of the decrease in a glutamate transporter in the context of ALS is unclear. Despite the fact that lower levels of EAAT2 have been reported in ALS, treatment with a compound that increases the expression of the EAAT2 transporter (ceftriaxone) had no therapeutic effect in ALS patients (Cudkowicz et al., 2014; not cited in this manuscript). No mutations have been found in EAAT2 and the down-regulation during the disease and in ALS models could be a consequence of the disease process, rather than a major cause of the selective motor neuron death.b) It is far from clear whether the phenotype observed in the Drosophila larvae has anything to do with ALS. The most important hallmark of ALS, selective motor neuron death, is not observed. It is not clear whether changes in the number and size of synaptic boutons, increases in the evoked excitatory junctional potentials and overexcitation of the motor neurons have anything to do with what happens during ALS. Only the motor deficits seem to be in line with this disease but it needs to be proven whether the cause of these motor defect is in any way related to what happens in ALS patients.c) Suggesting that antioxidants could be used as a therapeutic strategy for ALS is not very original and has been suggested many times before. Several antioxidants have been extensively tested in ALS without any success. The potassium channel blocker (4-AP) was also suggested before as a therapeutic strategy. However, this was mainly based on the hypoexcitability phenotype observed in motor neurons differentiated from iPSCs obtained from familial ALS patients. This defect was cell autonomous and had no direct relation with excitotoxicity and/or the interaction with other cell types.It is critical to reframe the revised manuscript without the strenuous connection to ALS. This will require major rewriting, but we am confident this is feasible within the revision period.

We really thank the reviewers for their suggestions and are happy that the reviewers find our work very exciting. We have significantly rewritten our manuscript with a major focus on the impact of the excitotoxicity-induced ROS on the integrity of the motor system.

Accordingly, we have modified the title and have rewritten the Abstract, Introduction, and Discussion. We have also added a Discussion part about potential relevance of ROS-induced motor circuit dysregulation for neurodegenerative diseases, in which we have cited and discussed this paper (Cudkowicz et al., 2014). The statement is in the Discussion section (subsection “Potential relevance of ROS-induced motor circuit dysregulation for neurodegenerative diseases”).

Finally, we have carried out the requested experiments and have added the figures of new data and the data statement to the “Results”.

Other important points:2) Can the authors use an independent eaat1 mutant or a VNC glia-specific knockdown of eaat1 and test it in some of the assays to show independent verification of the phenotype.

As requested by this reviewer, we have first expressed *UAS-eaat1-RNAi* (obtained from Dr. Donald van Meyel’s lab) in glia by using *repo-GAL4*. However, low knockdown efficiency of this RNAi transgene led to nearly normal locomotion. Therefore, we have further phenotypically characterized another independent mutant, *eaat1^hypo/SM2^*. The SM2 allele completely deletes *eaat1* gene (Stacey et al., 2010). *eaat1^hypo/SM2^*mutants displayed less Eaat1 expression compared to *eaat1^hypo/hypo^*mutants. This mutant mostly died between 1^st^ and 2^nd^instar larval stage, and ~ 1% of the mutants could further develop until 3^rd^instar stage, but they had a significant developmental delay for ~ 10 days. These data have been added in Figure 1—figure supplement 1C-E. The statement is in the text (subsection “Loss of *Drosophila eaat1* causes motor system deficits”, first paragraph).

We further phenotypically characterized *eaat1^hypo/SM2^* mutants that survived to the third instar stage. These larvae also exhibited alterations in locomotor CPG activity and locomotion. In addition, they presented outgrown NMJ boutons. However, *eaat1^hypo/SM2^* mutant NMJs had numerous satellite boutons and reduced muscle size, which are outcomes distinct from those found in *eaat1^hypo/hypo^* mutants. We speculate that this morphological difference may be partly attributable to a significant developmental delay in NMJ bouton growth (Sandoval et al., 2014). Furthermore, glial expression of *eaat1-venus* using *repo-GAL4* robustly corrected these defects.

Hence, these results strengthen the causal role of the *eaat1* mutation in altering the integrity of the motor system. These data have been added in Figure 2—figure supplement 3. The statement is in the text (subsection “Loss of *Drosophila* Eaat1 in astrocyte-like glia causes motor system defects”, last paragraph).

In addition, Figure 2D needs a neuronal control.

As requested by this reviewer, we have expressed the *eaat1-venus* transgene in neurons in *eaat1^hypo/hypo^* mutants. Interestingly, when we also tested the effect of neuronal expression of *eaat1-venus*, altered locomotor CPG output activity and NMJ boutons but not impaired locomotion were rescued, suggesting that ectopic expression of Eaat1 in neurons can partially restore function, but appropriate expression of Eaat1 in astrocytes is required to coordinate CPG activity between and/or within individual segments. These data have been added in Figure 2D-I and Figure 1—figure supplement 3A-B. The statement is in the text (subsection “Loss of *Drosophila* Eaat1 in astrocyte-like glia causes motor system defects”, second paragraph).

3) Discussion, fourth paragraph: The authors write that they find impaired muscle contraction caused by excess ROS exacerbates premotor circuit dysregulation. They showed only muscle contraction weakness in animals with reduced dSOD in the muscle. However, no data for premotor circuit dysregulation or increased ROS generation is shown. Please include also other phenotypes (burst activity, quantal content, ROS sensor) for this mutant condition.

As requested by this reviewer, we have expressed *UAS-dsod1-RNAi* in muscles using *C57-GAL4* and have measured burst activity, quantal content, and ROS level. We found that this manipulation specifically elevated ROS production in muscles but not in other tissues, such as the VNC. However, *dsod1* knockdown did not affect normal synaptic transmission or locomotor CPG activity. Therefore, ROS-induced muscle weakness acts to facilitate ROS accumulation of the locomotor CPG circuit and locomotor CPG dysfunction. These data have been added in Figure 5F and Figure 6—figure supplement 1C-I. The statement is in the text (subsection “ROS-induced muscle weakness feedback exacerbates premotor circuit dysfunction upon loss of *eaat1*”, second paragraph).

4) The authors claim that ROS generation in cholinergic neurons leads to attenuation of K^+^ channel inactivation, resulting in hypoexcitability. Blocking potassium channels by 4-AP rescues burst phenotypes but also rescues the increased ROS phenotype (Figure 4B). How can the blocking of K^+^ channels result in reduction of ROS caused by glutamate excitotoxicity? Is this via the feedback loop? This could be tested by specifically reducing dSOD in the muscle and treating these animals with 4-AP. This should not rescue ROS in the CPG.

We thank this reviewer for raising this issue. Based on our data, we think that the rescue effect by 4-AP treatment is indeed via the feedback loop. We did muscular knockdown of *dsod1*, also requested by above reviewer. However, under this condition, excess ROS production only occurs in muscles but not in the VNC, indicating that ROS-induced muscle weakness acts to facilitate CPG dysregulation. These data have been added in Figure 5F and Figure 6—figure supplement 1C-I. The statement is in the text (subsection “ROS-induced muscle weakness feedback exacerbates premotor circuit dysfunction upon loss of *eaat1*”, second paragraph).

Furthermore, we have asked if the feedback loop affects perisynaptic glutamate levels. We therefore treated *eaat1* mutants with 2 mM 4-AP to re-activate cholinergic interneurons, which reversed aberrant locomotor CPG activity and defective locomotion (Figure 5D-E). Consistent with the effect of muscular expression of hSOD1, the ROS increase in the *eaat1* mutant VNC was significantly relieved by means of 2 mM 4-AP treatment (Figure 4B). Interestingly, under this condition, the increase of perisynaptic glutamate in *eaat1* mutants was not affected (Figure 6M). A similar result was obtained by treatment with the antioxidant AD4 (Figure 6M). Hence, these results indicate that excitotoxicity can induce oxidative stress in the locomotor CPG circuit and muscles and lead to their dysfunction via a motor circuit-dependent mechanism, and the feedback arising from inefficient muscle contraction sustains oxidative stress in the locomotor CPG circuit independently of glutamate release. These data have been added in Figure 6M. The statement is in the text (subsection “ROS-induced muscle weakness feedback exacerbates premotor circuit dysfunction upon loss of *eaat1*”, last paragraph).

5) Neuronally expressing hSOD1 can fully suppress muscle ROS and muscle-expression of hSOD1 can fully suppress neuronal ROS. Can the author explain this?

According to our results, we propose a model (Figure 8) that glutamate excitotoxicity arising from the mutation in *Drosophila eaat1* should initiate low levels of ROS to slightly inactivate cholinergic transmission and prolong CPG output onto motor neurons. Subsequently, tonic premotor stimulation triggers ROS overproduction in muscles and dampens muscle contractility and consequent sensory input back to the locomotor CPG circuit, with this feedback strengthening the ROS increase within the CPG network to establish circuit dysregulation. In this scenario, neuronal expression of hSOD1 reversed the locomotor CPG defect and hence excess motor neuron stimulation to muscles and muscular oxidative stress. Although the results show a full suppression for excess ROS when hSOD1 was also expressed in muscles, we speculate that slight increase of ROS may still remain in the VNC, which likely is below the sensitivity of CMH2DCFDA. Since this is a feedback loop, reducing ROS at any point in this loop may prevent the feedback. Our non-autonomous rescue results therefore support the feedback hypothesis.

6) Several reviewers also found that the quality of presentation of the data could be improved.

We apologize for this language issue. As requested, the revised manuscript has been edited by native English editor. We hope that our new version is significantly improved and more transparent.